# DNMT and HDAC inhibition induces immunogenic neoantigens from human endogenous retroviral element-derived transcripts

Ashish Goyal[1,15], Jens Bauer [2,3,4,15], Joschka Hey [1,5,6,15], Dimitris N. Papageorgiou[7,8], Ekaterina Stepanova[9], Michael Daskalakis [1,14], Jonas Scheid [2,3,4,10], Marissa Dubbelaar [2,3,4,10], Boris Klimovich[4,11], Dominic Schwarz [7], Melanie Märklin [4,11], Malte Roerden[2,4], Yu-Yu Lin[1], Tobias Ma [12], Oliver Mücke[1], Hans-Georg Rammensee[3,4,13], Michael Lübbert [12], Fabricio Loayza-Puch [9], Jeroen Krijgsveld [7,8], Juliane S. Walz [2,3,4,11,16] ✉ & Christoph Plass [1,6,13,16] ✉

Immunotherapies targeting cancer-specific neoantigens have revolutionized the treatment of cancer patients. Recent evidence suggests that epigenetic therapies synergize with immunotherapies, mediated by the de-repression of endogenous retroviral element (ERV)-encoded promoters, and the initiation of transcription. Here, we use deep RNA sequencing from cancer cell lines treated with DNA methyltransferase inhibitor (DNMTi) and/or Histone deacetylase inhibitor (HDACi), to assemble a de novo transcriptome and identify several thousand ERV-derived, treatment-induced novel polyadenylated transcripts (TINPATs). Using immunopeptidomics, we demonstrate the human leukocyte antigen (HLA) presentation of 45 spectra-validated treatment-induced neo-peptides (t-neopeptides) arising from TINPATs. We illustrate the potential of the identified t-neopeptides to elicit a T-cell response to effectively target cancer cells. We further verify the presence of t-neopeptides in AML patient samples after in vivo treatment with the DNMT inhibitor Decitabine. Our findings highlight the potential of ERV-derived neoantigens in epigenetic and immune therapies.

Anti-cancer T cell responses play a major role in the immune surveillance of malignant disease. Effective T cell-based cancer immune control requires functional and tumor-specific T cells as well as the presentation of antigen targets on human leukocyte antigen (HLA) molecules on the tumor cell surface. Such tumor antigens can arise from proteins with differential expression, processing, or HLA presentation in the tumor cells (e.g. cancer-testis antigens) as well as from tumor-specific mutations or non-canonical gene products, de novo

expressed in tumor cells[1–5]. Whereas several anti-cancer drugs have been developed to induce priming or reinvigoration of cancer-specific effector T cells, including immune checkpoint inhibitors (ICI) and immunomodulatory drugs, so far only little is known about how cancer drugs affect the target structures of anti-cancer T cell responses and thus tumor antigen presentation[6–8].

Epigenetic therapies, alone or in combination with immunotherapeutic approaches (ICI therapy, donor lymphocyte infusion (DLI))

---

have emerged as promising avenues for the treatment of human malignancies which are currently tested in clinical trials[9–13]. Three routes of antigen target-based immune modulation mediated by the DNA methyltransferase inhibitors (DNMTi), 5-azacytidine (AZA), or 5-aza-2′-deoxycytidine (DAC), have been postulated. These include (i) the induction of Cancer Testis Antigens (CTAs) through promoter demethylation and (ii) the activation of Human Endogenous Retroviral Elements (HERV), triggering the cellular Interferon-alpha (INF-α) response through the dsRNA sensing mechanism, thereby inducing viral mimicry[14–17]. As a third mechanism (iii), we previously proposed that DAC treatment, alone or in combination with histone deacetylase inhibitors (HDACi), induces solitary HERVs, mainly of the long terminal repeat (LTR) 12C subfamily (LTR12C)[18]. The activation of the LTR12C promoter elements leads to treatment-induced non-annotated transcription start site-induced transcripts. These treatment-induced transcripts can splice into protein-coding exons and encode for truncated and chimeric open reading frames (ORFs) that were hypothesized to encode for putative neoantigens, which are displayed on HLA[18]. However, the existence of these neoantigens has never been demonstrated experimentally.

In this study, we identify that DAC alone or in combination with HDACi induces novel, full-length polyadenylated transcripts. We characterize these transcripts by deep RNA sequencing and de novo transcriptome assembly, and demonstrate the existence of immunogenic t-neopeptides via immunopeptidomics, both in vitro and in vivo. Utilizing a cancer cell line panel, we show that the de-repression of LTRs drives transcription and is universally found across multiple cancer entities upon DAC and HDACi treatment. Furthermore, we use this comprehensive de novo assembly of the cancer cell line panel to identify the induction of t-neopeptides in acute myeloid leukemia (AML) patients treated with DAC. In summary, we describe a repertoire of epigenetic inhibitor-induced neoantigens that can be further exploited for immune therapy.

## Results

### De novo transcriptome assembly identifies full-length, treatment-induced, novel polyadenylated transcripts

In our previous work, we utilized Cap Analysis Gene Expression sequencing (CAGE-seq) and identified 2,362 non-annotated transcription start sites in DAC and DAC/HDACi treated NCI-H1299 (human non-small cell lung carcinoma) cells[18]. CAGE-seq does not discriminate between polyAdenylated (polyA+) and non-polyAdenylated (polyA−) transcripts, targets the 5′ end of transcripts, and therefore, does not determine the full-length sequence of long polyA+ transcripts[19]. Thus, the data in our previous work is lacking crucial information which is needed to predict open reading frames (ORFs) and the coding capacity of novel transcripts. To annotate treatment-induced novel polyA+ transcripts (TINPATs), we performed deep sequencing of polyA+ RNA (RNA-seq) from DMSO-, DAC-, SB939-, and DAC + SB939-treated NCI-H1299 cells (Fig. 1a upper panel). Using deep RNA-seq, we obtained improved sequencing read coverage throughout the gene body of all known transcripts (Supplementary Fig. 1a, b), as exemplified by the GAPDH locus (Supplementary Fig. 1c). De novo transcriptome assembly (see methods, Supplementary Fig. 2a) generated from the entire RNA-seq dataset identified 207,411 known, GENECODE-annotated transcripts as well as 30,695 novel transcripts (Fig. 1b, Supplementary Data 1). In comparison to the GENECODE assembly, the de novo assembly achieved a precision of >95% and a sensitivity of >99% at the locus level (Supplementary Fig. 2b). Novel transcripts were further classified into transcripts that splice into known genes and form chimeric transcripts ($n = 26,866$) and transcripts not associated with known transcription (non-chimeric, $n = 3829$). Differential transcript expression analysis revealed that many transcripts underwent significant deregulation (adjusted (adj.) $p$-value < 0.01, absolute (abs.) $\log_2$ fold change > 2) upon DNMTi and/or HDACi treatment (Fig. 1c,

Supplementary Data 2). In line with previous observations, DAC + SB939 treatment resulted in the highest number of deregulated transcripts (6650 upregulated and 445 downregulated transcripts, Fig. 1c) and accounted for almost all of the upregulated transcripts across any treatment regimen (Supplementary Fig. 2c). Moreover, novel transcripts showed stronger induction upon DAC + SB939 treatment as compared to known transcripts and other treatment regimens (Fig. 1d). We further focused our analysis on DAC + SB939 differentially induced transcripts ($\log_2$ fold change > 2, adj.$p$-value < 0.01) and identified 3023 TINPATs (Fig. 1e). 703 TINPATs were found to arise from previously published treatment-induced non-annotated transcription start sites (Supplementary Fig. 2d). From the remaining 1642 treatment-induced non-annotated transcription start sites we could not detect any full-length polyA+ transcripts, possibly due to low expression levels. The chimeric TINPATs were longer (average length ~69 kbps) and had more exons as compared to treatment-induced GENECODE-annotated transcripts (average length ~49 kbps). In contrast, non-chimeric transcripts were shorter (average length ~19 kbps) and had fewer exons, with 35% of all non-chimeric transcripts being mono-exonic (Fig. 1f, g). As expected, TINPAT expression in normal tissues (accessed in ENCODE data) was very low to absent with the exception of the testis, which showed basal expression of TINPATs (Supplementary Fig. 2e, Supplementary Data 3). Most of the chimeric TINPATs were predicted to have high coding potential, whereas the non-chimeric TINPATs were predicted to have a low protein-coding potential (Supplementary Fig. 2f). Previously, we demonstrated that treatment-induced novel transcription often occurs from transposable elements (TEs)[18]. In contrast to known DAC + SB939 treatment-induced transcripts, the majority of TINPATs are initiated from TEs with 45% of chimeric TINPATs and 70% of non-chimeric transcripts arising from LTR repeats (Fig. 1h). Locus overlap and enrichment analysis (LOLA) revealed the ERV1 class and in particular, the LTR family, and the LTR12C subfamily as the major source of TINPATs, confirming previous results of LTR12C elements serving as promoters upon epigenetic drug treatment[18] (Fig. 1i, Supplementary Fig. 2g). Overall using RNA-seq, we obtained superior sequencing read coverage over CAGE-seq throughout the known, novel non-chimeric and chimeric transcripts as shown by the chimeric TCN2 (Fig. 1j) and the non-chimeric MSTRG.32198 transcripts (Fig. 1k).

### Novel ORF-transcripts as a source of HLA-presented neoantigens after DAC + SB939 treatment of NCI-H1299 cells

Using this comprehensive de novo transcriptome assembly, we predicted all possible ORFs from DAC + SB939-induced transcripts with a size greater than seven amino acids (Fig. 2a, Supplementary Data 4). Non-chimeric TINPATs were predicted to encode smaller (average ORF length = 29.2) and lower number of ORFs per transcript (average = 18) as compared to known induced transcripts (average ORF length = 50.3 and average number of ORFs per transcript = 20). Chimeric TINPATs were predicted to encode slightly smaller ORFs (average ORF length = 41.5) and a higher number of ORFs per transcript (average = 24) as compared to known induced transcripts (Fig. 2b, c, Supplementary Fig. 3a). To identify treatment-induced proteins, we used an azidohomoalanine (AHA) pulse SILAC labeling approach[20]. AHA is a methionine analog that bears an azide group and can be readily incorporated into the newly synthesized proteins. Using alkyne beads the newly synthesized proteins are immobilized onto the beads via click chemistry. The combination of this enrichment approach with SILAC labeling allowed us to capture the proteins that are newly translated upon the administration of DAC + SB939 treatment compared to the DMSO control. This methodology enabled the study of the proteins/ peptides that are specifically de novo synthesized in response to the DAC + SB939 treatment. Whole-cell proteomics analysis identified 272 proteins from these predicted ORFs, confirming the protein-coding capacity

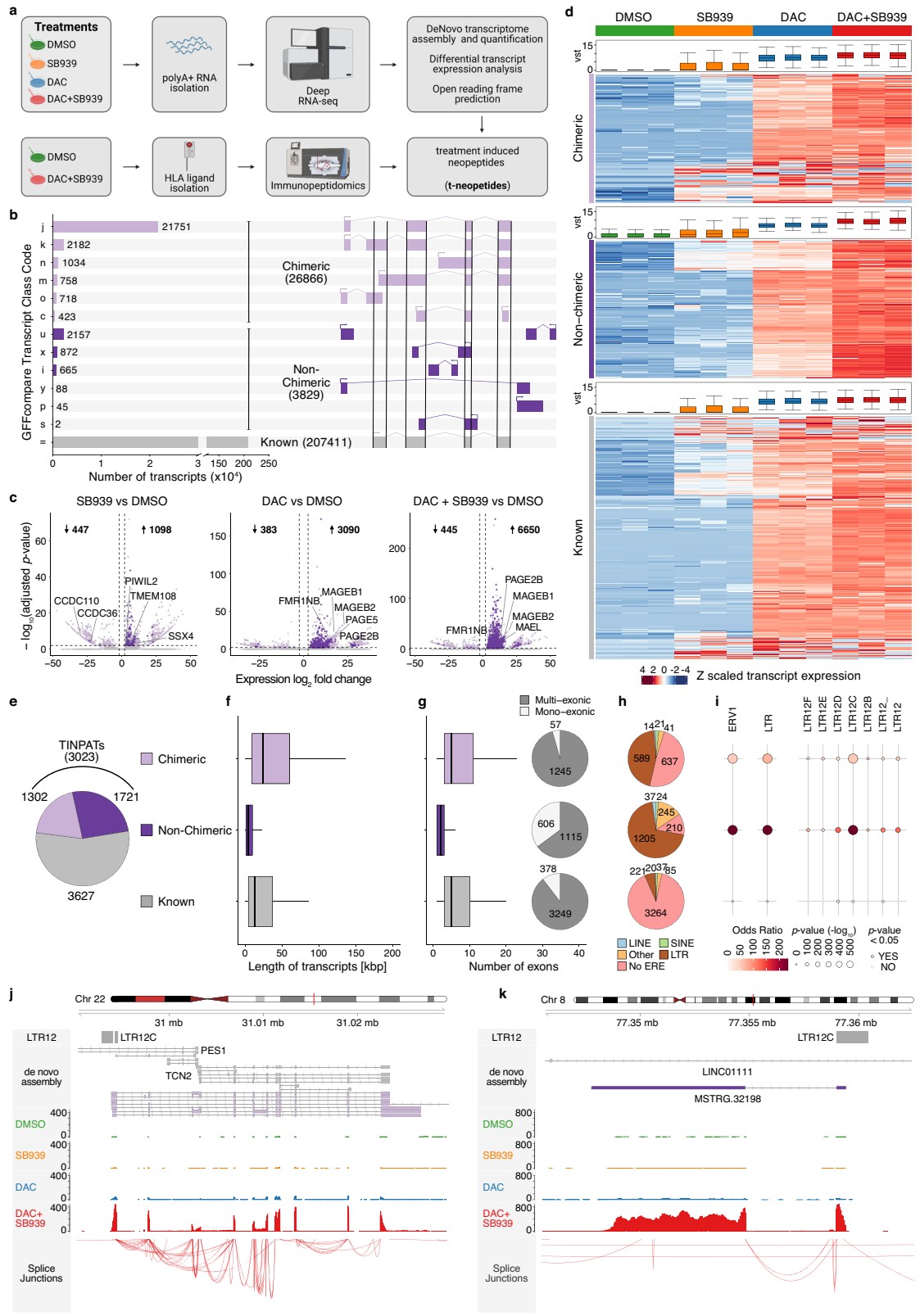

of these novel ORFs (Fig. 2d, Supplementary Fig. 3b, Supplementary Data 5).

To identify novel, treatment-induced ORF-derived Human Leukocyte Antigen (HLA)-presented neopeptides (t-neopeptides), we analyzed HLA class I-presented peptides of NCI-H1299 cells after DAC + SB939 treatment using liquid chromatography-coupled tandem mass spectrometry (LC-MS/MS) in comparison to DMSO-treated

control cells (Supplementary Fig. 3c). MS-based immunopeptidome analysis revealed 5264 and 5487 different HLA class I ligands after DAC + SB939 treatment or in the DMSO control, respectively (Fig. 2e, Supplementary Data 6). Overlap analysis showed a high similarity of all ligands identified in the immunopeptidome of DAC + SB939 and DMSO-treated controls (Fig. 2f, left). A total of 112 HLA ligands were exclusively assigned to the novel ORF repertoire, composed of

**Fig. 1 | De novo transcriptome assembly identifies treatment-induced novel poly-adenylated transcripts (TINPATs). a** Schematic overview of the experimental setup comprising the de novo transcriptome assembly of DMSO, Decitabine (DAC), SB939, or DAC + SB939-treated NCI-H1299 cells as well as the isolation of (human leukocyte antigen) HLA-presented ligands from DMSO or DAC + SB939-treated NCI-H1299 cells followed by (mass spectrometry) MS-based immunopeptidome analysis (created with BioRender). **b** RNA-seq performed on biological replicates (*n* = 3) of NCI-H1299 cells treated with DMSO, SB939, DAC, or DAC + SB939 was used to generate a de novo transcriptome assembly (see methods). **c** Volcano plot of differential transcript expression (defined by DESeq2: adjusted *p*-value < 0.01, absolute log$_2$ fold change > 2) analyses for indicated comparisons. The color of the dots indicates the transcript classification (gray: known, purple: non-chimeric, light purple: chimeric). **d** Heatmap of z-scaled variance stabilized transformed (vst) transcript expression of DAC + SB939-induced transcripts stratified by their transcript classification. Box plots of vst transcript expression are plotted alongside. **e**–**g** Number (**e**), length (**f**), and the number of exons (**g**) of

DAC + SB939-induced transcripts stratified by their transcript classification. The number of mono- and multi-exonic transcripts are plotted alongside. Box plots indicate the largest value within the 1.5 times interquartile range above 75$^{th}$ percentile, 75$^{th}$ percentile, median, 25$^{th}$ percentile, and smallest value within the 1.5 times interquartile range below 25$^{th}$ percentile (*n* = 3 biological replicates). **h** Number of transcriptional start sites (TSSs) of DAC + SB939-induced transcripts, overlapping with transposable element (TE) families and their (**i**) locus overlap and enrichment analysis with TE families, subfamilies, and classes stratified by transcript classification. The most significant enrichment results for TE families and subfamilies, and LTR12-derived classes, are visualized. TSSs of all identified transcripts served as background (*p*-values were calculated using a Fisher's test as the statistical framework within R package LOLA). **j**, **k** Locus plots of selected chimeric (**j**) and non-chimeric (**k**) novel transcripts, showing (from top to bottom) LTR12 repeats, the de novo transcriptome assembly, normalized RNA-seq coverage, splicing sites of DAC + SB939 treated NCI-H1299 cells.

TINPAT-derived ORFs and non-canonical ORFs from known transcripts with 63 treatment-specific HLA ligands (Fig. 2f, right panel, Supplementary Data 7). Novel ORF-derived HLA class I ligands showed a characteristic and similar length distribution compared to canonical HLA ligands (73.2% and 74.0% 9mers, respectively; Fig. 2g) as well as intensity ranks covering the whole range of the immunopeptidome ligand abundance (Fig. 2h). 56% (63) of the DAC + SB939-induced t-neopeptides were identified exclusively after DAC + SB939 treatment with 15 t-neopeptides identified in all three replicates of the immunopeptidome analysis (Fig. 2i, Table 1). Comparative spectra analysis using synthetic peptides and calculation of the spectral correlation coefficients ($R^2$) validated 70/112 (62.5%, $R^2 \geq 0.70$) t-neopeptides, comprising the 15 treatment-exclusive t-neopeptides identified in all three replicates of the immunopeptidome (Fig. 2i, j, Supplementary Fig. 3d, e). The calculated $R^2$ values showed a direct dependency on the sample identification frequency indicating the importance to focus on highly frequent treatment-exclusive candidates for the selection of relevant t-neopeptides (Fig. 2k). Post-translational modification (PTM)-analysis revealed no treatment-induced change in the frequency of PTMs (Supplementary Fig. 3f).

66% of the t-neopeptides, identified in at least two out of three replicates of the immunopeptidome analysis (*n* = 38), arose from a unique transcript (*n* = 25). The remaining 34% were predicted to arise from multiple transcripts (Fig. 3a, Supplementary data 8) Notably, one of the t-neopeptides was predicted to arise from ORFs encoded by 88 transcripts (Supplementary Fig. 4a); the corresponding DNA sequence encoding for this t-neopeptide lies within the conserved LTR12C sequence (exemplified in Supplementary Fig. 4b). All other t-neopeptides were encoded by non-LTR12C sequences (exemplified in Fig. 3b). TINPATs were found to be the major source (*n* = 146) of t-neopeptides with a tiny fraction encoded by non-canonical ORFs lying within known transcripts (*n* = 8) (Fig. 3c). Despite their low predicted protein-coding potential, non-chimeric TINPATs (*n* = 106) accounted for the majority of TINPAT-derived t-neopeptides (Supplementary Fig. 2f). t-neopeptides encoding transcripts were mostly derived from promoters encoded in LTR repeats of the LTR12C subfamily (Fig. 3d, e). As expected, they were strongly induced upon DAC + SB939 treatments. Yet, these transcripts also showed a strong induction upon DAC treatment alone, indicating that DAC treatment could be sufficient to induce t-neopeptides (Fig. 3f). To investigate if these t-neopeptides arose from homologous regions elsewhere in the genome, we blasted their peptide sequences against all known human proteins and all predicted DAC + SB939-induced ORFs, allowing for 1, 2, or 3 amino acid mismatches. 23/38 (61%), or 16/38 (42%) of the t-neopeptides were uniquely mapped allowing for 1, or 2 mismatches, respectively (Supplementary Fig. 4c, d, Supplementary data 8). The multi-mapping t-neopeptides were mainly derived from LTR12-derived transcripts (Supplementary Fig. 4e, f, Supplementary data 8). To assess

the impact of sequence polymorphism in human genomes on t-neopeptides, we determined the percentage of polymorphism at DNA, as well as amino acid level for all the t-neopeptides in Table 1. All of the t-neopeptides had a very low sequence polymorphism (Supplementary data 9).

**Identification and characterization of novel ORF-specific CD8$^+$ T cells**

To investigate the immunogenicity of treatment-induced novel ORF-derived HLA ligands, in vitro priming of peptide-specific cytotoxic T lymphocytes was performed using the HLA-A*32:01 and the HLA-A*24:02 assigned t-neopeptides LLLSYRYIY (P$_{A*32}$, SYFPEITHI score of 72.7% and NetMHCpan rank of 0.841) and SYLKYVFQL (P$_{A*24}$, SYFPEITHI score of 80.7% and NetMHCpan rank of 0.012), respectively. Artificial antigen-presenting cell (aAPC)-based priming of CD8$^+$ T cells of healthy volunteers (HVs, *n* = 3) with matching HLA allotypes (Supplementary Table S1) showed induction of P$_{A*32}$- and P$_{A*24}$-specific CD8$^+$ T cells with frequencies of up to 29.1% (median 4.9%; Fig. 4a, b) and 12.7% (median 2.6%; Supplementary Fig. 5a, b) of peptide-specific T cells, respectively. P$_{A*32}$- and P$_{A*24}$-specific CD8$^+$ T cells showed a polyfunctional phenotype reflected by specific IFN-γ and TNF production upon target t-neopeptide stimulation (Fig. 4c, Supplementary Fig. 5c). In addition, P$_{A*32}$-specific CD8$^+$ T cells showed specific lysis of DAC + SB939-treated NCI-H1299 cells in vitro (Fig. 4d), validating the cytotoxic T cell activation upon treatment-induced novel ORF-derived HLA ligand presentation.

**The induction of treatment-induced transcripts is conserved across different cancer entities**

To demonstrate that TINPAT induction upon DAC and HDACi treatment is not restricted to NCI-H1299 cells, we performed DNMT and HDAC inhibition in a large panel of cancer cell lines of several tissue types and cancer entities, including lung cancer, acute myeloid leukemia, glioblastoma, and colon cancer. Since LTR12C was the most significantly induced repeat family in NCI-H1299 treated cells, we performed a qRT-PCR using primers to detect global LTR12C-derived transcription using primers in conserved regions of LTR12C. We observed a strong induction of LTR12C expression upon DAC or SB939 and in particular for the double-treatment for all cell lines (Fig. 5a). Furthermore, we utilized RNA-seq on a subset of cell lines to generate a pan-cancer treatment-specific de novo transcriptome assembly (Supplementary Data 10). Differential transposable element expression analysis revealed the LTR12 family as the most strongly induced repeat family across all cancer cell lines (Supplementary Fig. 6a, b). Principal component (PC) analysis (PCA) of the 5000 most variably expressed transcripts of the de novo assembly showed that the majority of variation (13.06%) was caused by the drug treatment, clearly separating DAC + SB939 from DMSO-treated

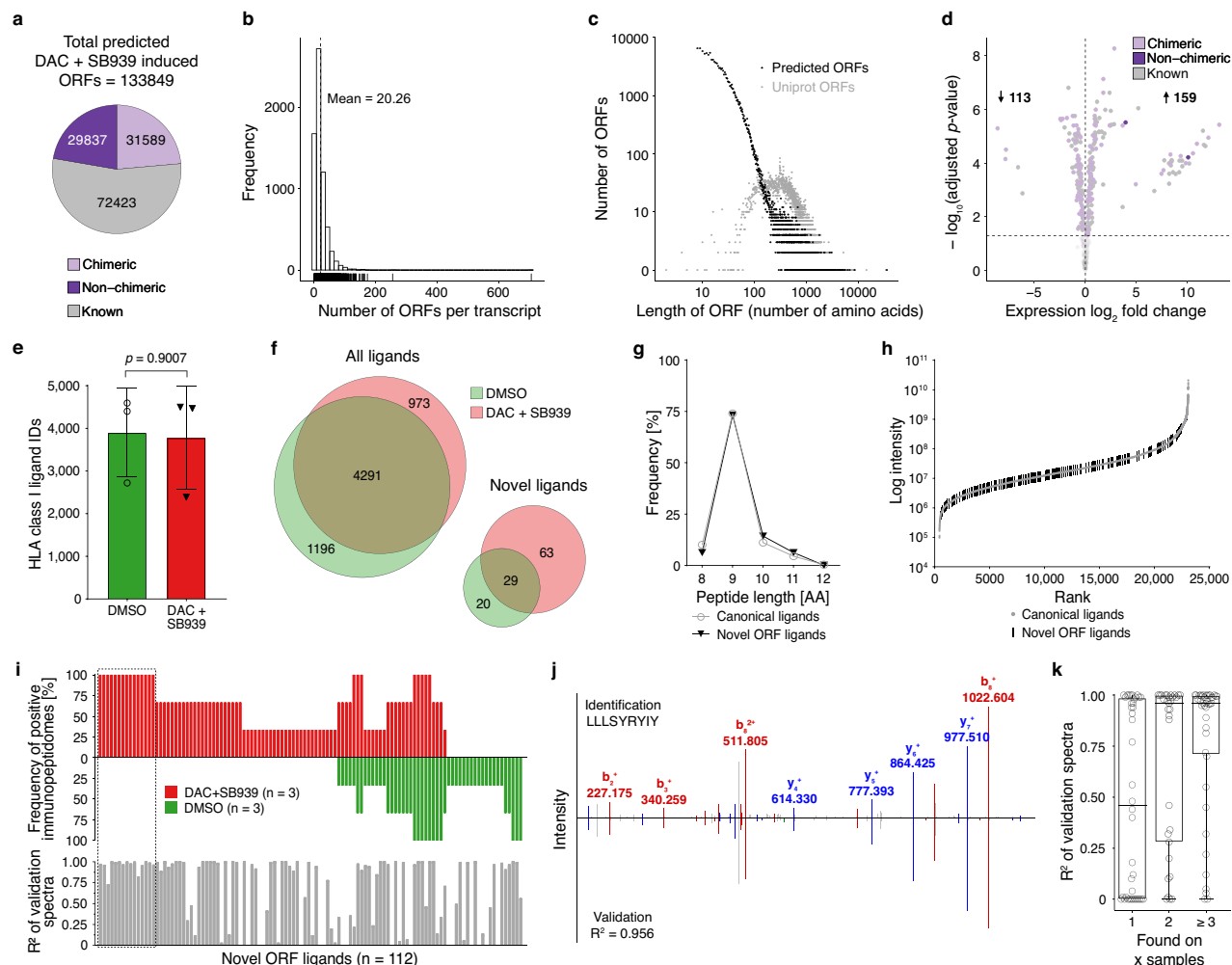

**Fig. 2 | Immunopeptidomics unravels DAC + SB939-induced novel ORFs-derived HLA ligands. a** Number of ORFs predicted from DAC + SB939-induced transcripts stratified by their host transcript classification. **b** Frequency distribution of the number and (**c**) length of the predicted ORFs. **d** Volcano plot of differential protein abundance analyses, comparing DAC + SB939 vs DMSO treated NCI-H1299 cells measured by MS using predicted ORFs from DAC + SB939-induced transcripts as reference. **e** MS-identified HLA class I ligands from DAC + SB939 vs DMSO treated cells ($n = 3$ biological replicates; mean values ± SD; $p$-value was calculated using an unpaired two-tailed $t$ test). **f** Overlap analysis of MS-identified HLA ligands from DAC + SB939 vs DMSO treated cells from all identified ligands (left) or novel ORF-exclusive ligands (right). **g** Peptide length distribution of HLA ligands exclusively assigned to novel ORFs compared to all identified canonical HLA ligands. **h** Ranked intensity values of MS acquired data derived from the combined immunopeptidomes of DAC + SB939 and DMSO treated cells ($n = 3$). Positions of novel ORF-derived HLA ligands are projected on the curve. **i** Comparative profiling of ORF-derived HLA ligands ($n = 112$) based on HLA-restricted presentation frequency in immunopeptidomes of DAC + SB939 or DMSO treated cells. Frequencies of positive immunopeptidomes for the respective HLA ligands ($x$-axis) are indicated on the $y$-axis. The box highlights treatment-induced neopeptides (t-neopeptides; $n = 15$) with presentation in 100% of the biological replicates. The corresponding calculated spectral correlation coefficients ($R^2$) of the fragment spectra comparison are indicated below. **j** Fragment spectra comparison ($m/z$ on the $x$-axis) of the experimentally eluted DAC + SB939-induced novel ORF-derived HLA class I-presented ligand LLLSYRYIY ($P_{A*32}$) extracted from the DAC + SB939 treated cells (identification) to the synthetic peptide (validation, mirrored on the $x$-axis) with the calculated $R^2$. Identified b- and y-ions are marked in red and blue, respectively. **k** Calculated $R^2$ of the fragment spectra comparison of experimentally eluted and synthetic peptides plotted against the number of samples with ligand identification ($n = 6$). The band indicates the median, and the box indicates the first and third quartiles.

samples in PC1 (Fig. 5b, Supplementary Fig. 6c). Overall, we identified 42,580 additional transcripts as compared to the NCI-H1299 de novo assembly (Supplementary Fig. 6d). Differential transcript expression analysis uncovered 14,957 differentially induced transcripts of which 9369 were TINPATs (Fig. 5c, Supplementary Data 11). The majority of TINPATs showed strong induction levels across all cancer cell lines (Fig. 5d).

### Identification of t-neopeptides in AML patients treated in vivo with DAC

To investigate the clinical relevance of t-neopeptides we analyzed HLA class I- and HLA class II-presented peptides by LC-MS/MS in blood samples from two AML patients before and 48/96 h after in vivo DAC treatment (Fig. 6a, Supplementary Data 12). Over all time points, a total of 10,563 and 6641 unique HLA class I- and HLA class II-presented peptides were identified (Fig. 6b, c). A database search applying the predicted ORFs from the pan-cancer treatment-specific de novo transcriptome assembly revealed a total of 124 unique ORF-derived HLA class I-presented ligands in the immunopeptidomes of the two AML patients. 70% ($n = 87$) of these ligands were exclusively presented after DAC treatment, representing in vivo induced t-neopeptides (Fig. 6d, Supplementary Data 13). For HLA class II, 29 unique ORF-derived peptides were identified, with 69% ($n = 20$) exclusively presented after DAC treatment (Fig. 6e, Supplementary Data 13). aAPC-based priming of CD8+ T cells of HLA-matched HVs with the in vivo identified HLA class I t-neopeptides RTDSSLLEK (A*03:01; $P_{A*03}$), TPSLRTVTL (B*07:02; $P_{B*07-1}$), and NPRPGLGAL (B*07:02; $P_{B*07-2}$) (Supplementary Table S1), showed induction of

**Table 1 | Characteristics of the validated t-neopeptides**

| t-neopeptide | q-Value | XCorr | HLA restriction SYFPEITHI | SYFPEITHI score [% of max. score] | HLA restriction NetMHC | NetMHCpan [rank] | Predicted source ORF | ORF length | Source transcript class | LTR12 derived |
|---|---|---|---|---|---|---|---|---|---|---|
| VMFQDFVSW | 0.000 | 3.102 | A*32:01 | 39.39 | A*32:01 | 0.014 | ORF_MSTRG.23152.1.p3 | 29 | Non-chimeric | LTR12C |
| LLLSYRYIY | 0.000 | 3.369 | A*32:01 | 72.73 | A*32:01 | 0.841 | ORF_MSTRG.28868.1.p28 | 12 | Non-chimeric | LTR12 |
| VLSVSRFSF | 0.000 | 2.822 | A*32:01 | 78.79 | A*32:01 | 0.177 | ORF_MSTRG.31576.1.p89 | 10 | Chimeric | NO |
| VTKSTSLPW | 0 | 3.22 | A*32:01 | 63.64 | A*32:01 | 0.095 | ORF_MSTRG.29327.1.p7 | 78 | Chimeric | NO |
| | | | | | | | ORF_ENST00000354536.9_3.p8 | 65 | Known | NO |
| SYLKYVFQL | 0.000 | 2.443 | A*24:02 | 80.65 | A*24:02 | 0.012 | ORF_MSTRG.29191.1.p4 | 28 | Non-chimeric | LTR12C |
| | | | | | | | ORF_MSTRG.29191.2.p6 | 28 | Non-chimeric | LTR12C |
| IIAQLLNSTW | 0.000 | 3.589 | A*32:01 | 63.89 | A*32:01 | 0.637 | ORF_MSTRG.28772.1.p3 | 24 | Non-chimeric | LTR12F |
| KEQTPDTPSL | 0 | 2.9 | B*40:01 | 68.75 | B*40:01 | 0.081 | ORF_MSTRG.7601.1.p1 | 44 | Non-chimeric | LTR12 |
| | | | | | | | ORF_MSTRG.16894.1.p1 | 173 | Chimeric | LTR12 |
| | | | | | | | ORF_MSTRG.16894.3.p1 | 182 | Non-chimeric | LTR12 |
| | | | | | | | ORF_MSTRG.19191.1.p2 | 76 | Non-chimeric | LTR12 |
| | | | | | | | ORF_MSTRG.25496.1.p1 | 103 | Non-chimeric | LTR12C |
| | | | | | | | ORF_MSTRG.25979.1.p1 | 83 | Non-chimeric | LTR12C |
| YTKLIGGYY | 0 | 3.81 | C*02:02 | 71.43 | C*02:02 | 0.141 | ORF_MSTRG.16351.4.p1 | 87 | Non-chimeric | LTR12D |
| | | | | | | | ORF_MSTRG.16351.5.p1 | 87 | Non-chimeric | LTR12D |
| RIFPVGLNSF | 0.000 | 3.511 | A*32:01 | 80.56 | A*32:01 | 0.008 | ORF_MSTRG.29412.1.p2 | 56 | Non-chimeric | LTR12C |
| RTAPFTSSY | 0.002 | 2.774 | A*32:01 | 84.85 | A*32:01 | 0.007 | ORF_ENST00000206423.7_2.p55 | 20 | Known | NO |
| LISGEGLLL | 0.007 | 2.2 | C*02:02 | 64.29 | C*03:03 | 1.016 | ORF_ENST00000409039.7_3.p35 | 23 | Known | NO |
| | | | | | | | ORF_ENST00000492261.5_1.p10 | 23 | Known | NO |
| RVADLVLIF | 0.002 | 1.899 | A*32:01 | 75.76 | A*32:01 | 0.006 | ORF_MSTRG.14951.2.p3 | 59 | Non-chimeric | LTR12C |
| SEEPLRPAA | 0.002 | 2.638 | B*40:02 | 42.42 | B*40:02 | 0.386 | ORF_MSTRG.7564.7.p10 | 16 | Non-chimeric | NO |
| LYGPQSNSFF | 0.000 | 3.268 | A*24:02 | 70.00 | A*24:02 | 0.065 | ORF_MSTRG.31594.19.p1 | 75 | Chimeric | NO |
| KLQSIIYIY | 0.023 | 2.198 | A*32:01 | 78.79 | A*32:01 | 0.037 | ORF_MSTRG.14414.2.p3 | 106 | Non-chimeric | LTR12C |

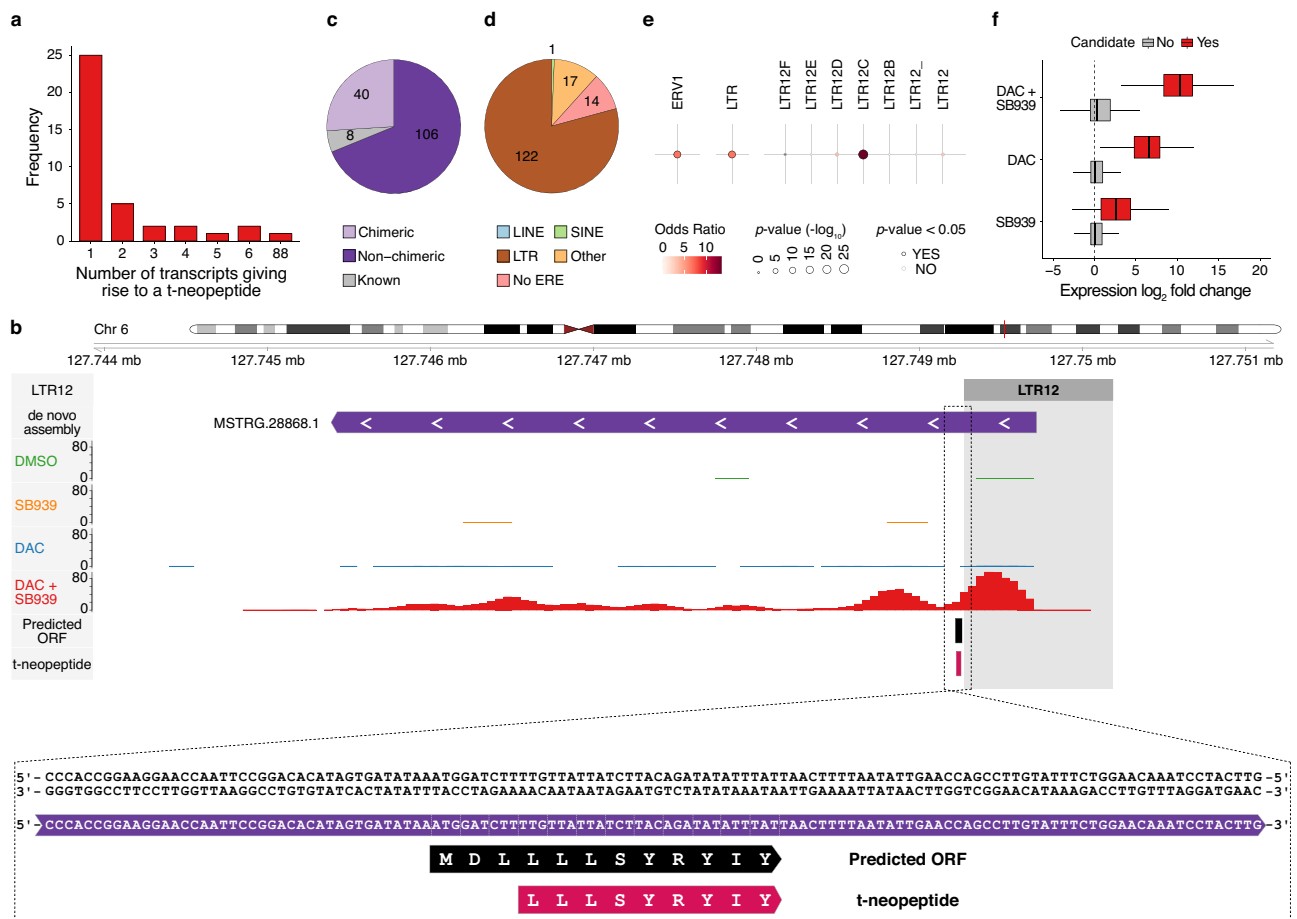

**Fig. 3 | t-neopeptide encoding transcripts are mostly LTR12-derived. a** The number of DAC + SB939-induced transcripts giving rise to peptides identified in more than one biological replicate of DAC + SB939 treated samples and absent in DMSO treated samples. **b** Locus plot of a selected treatment-induced novel polyA+ transcript (TINPAT) giving rise to t-neopeptides outside repeat sequence, showing (from top to bottom) LTRs, the de novo transcriptome assembly, normalized RNA-seq coverage of DMSO, SB939, DAC, and DAC + SB939 treated NCI-H1299 cells, predicted ORF, and corresponding t-neopeptide LLLSYRYIY. The region containing the t-neopeptide is zoomed in and reversed for improved visualization. **c** Classification of transcripts giving rise to t-neopeptides in known, chimeric, and non-chimeric novel transcripts. **d** The number of transcriptional start sites of transcripts that give rise to t-neopeptides and overlap with transposable element families. **e** Locus overlap and enrichment analysis of transcripts that give rise to t-neopeptides with transposable element families, subfamilies, and classes. The most significant enrichment result for transposable element families and subfamilies, as well as all LTR12-derived classes, are visualized. Transcriptional start sites of all DAC + SB939 treatment-induced transcripts were used as a background for enrichment analysis. *p*-values were calculated using Fisher's exact test. **f** Transcript expression fold changes of transcripts that give rise to t-neopeptides (red) and all transcripts (gray) in the DAC + SB939 vs DMSO, DAC vs DMSO, and SB939 vs DMSO comparison. Box plots indicate the largest value within the 1.5 times interquartile range above 75th percentile, 75th percentile, median, 25th percentile, and smallest value within the 1.5 times interquartile range below 25th percentile (*n* = 3 biological replicates).

P$_{A*03}$-specific CD8$^+$ T cells with a frequency of 0.8% (Supplementary Fig. 7a). Moreover, peptide-specific CD4$^+$ memory T cells targeting a panel of HLA class II presented t-neopeptides (*n* = 9) identified in the immunopeptidomics analysis of the AML patients, were identified in one out of the eight AML patients that had undergone DNMTi treatment in vivo (Fig. 6f, Supplementary Data 12, 13). Similar to the t-neopeptides identified in DAC + SB939-treated NCI-H1299 cells, HLA class I and II-presented ligands of AML patients 48/96 h post-DAC treatment, arose mostly from unique TINPATs, driven by ERV and, in particular, LTR12C-encoded promoter elements (Supplementary Fig. 7b–i). The ORF ORF_MSTRG.21956.5.p2, to which the HLA class I-ligand TPSLRTVTL was assigned, was also identified in the NCI-H1299 RNA-seq and Ribo-seq analysis and has direct sequence overlap with the NCI-H1299 t-neopeptide KEQTPDTPSL (Fig. 6f, Supplementary Fig. 7j). This highlights the effectiveness of the pan-cancer treatment-specific de novo transcriptome assembly approach for the identification of naturally presented t-neopeptides in patient samples that could be used for the development of novel combinatorial immunotherapy approaches.

## Discussion

T cell recognition of HLA-presented antigens plays a central role in the immune surveillance of malignant disease[21,22]. Numerous immunotherapeutic approaches aim to utilize respective tumor antigens to therapeutically induce an anti-tumor T cell response[23–25]. Induction of treatment-specific neoantigens by anticancer drugs might open novel avenues of combinatorial cancer immunotherapy approaches[18,26,27].

In this study, we report a de novo transcriptome assembly of NCI-H1299 cells following DMSO, DAC, SB939, or DAC + SB939 treatment to reliably predict novel ORFs encoded in treatment-induced transcripts. A total of 6650 differentially induced transcripts and 3,023 TINPATs were identified post-combination treatment, encoding for 61,426 predicted ORFs. The majority of TINPATs initiated from promoter sequences within LTR12 repeats, confirming our previous data[18]. In line with published work, we observed a strong activation of epigenetically silenced genes[20] as well as CTA coding genes[21], both upregulated upon DAC, SB939, or DAC + SB939 treatment (Supplementary Fig. 8a–f, Supplementary Data 14).

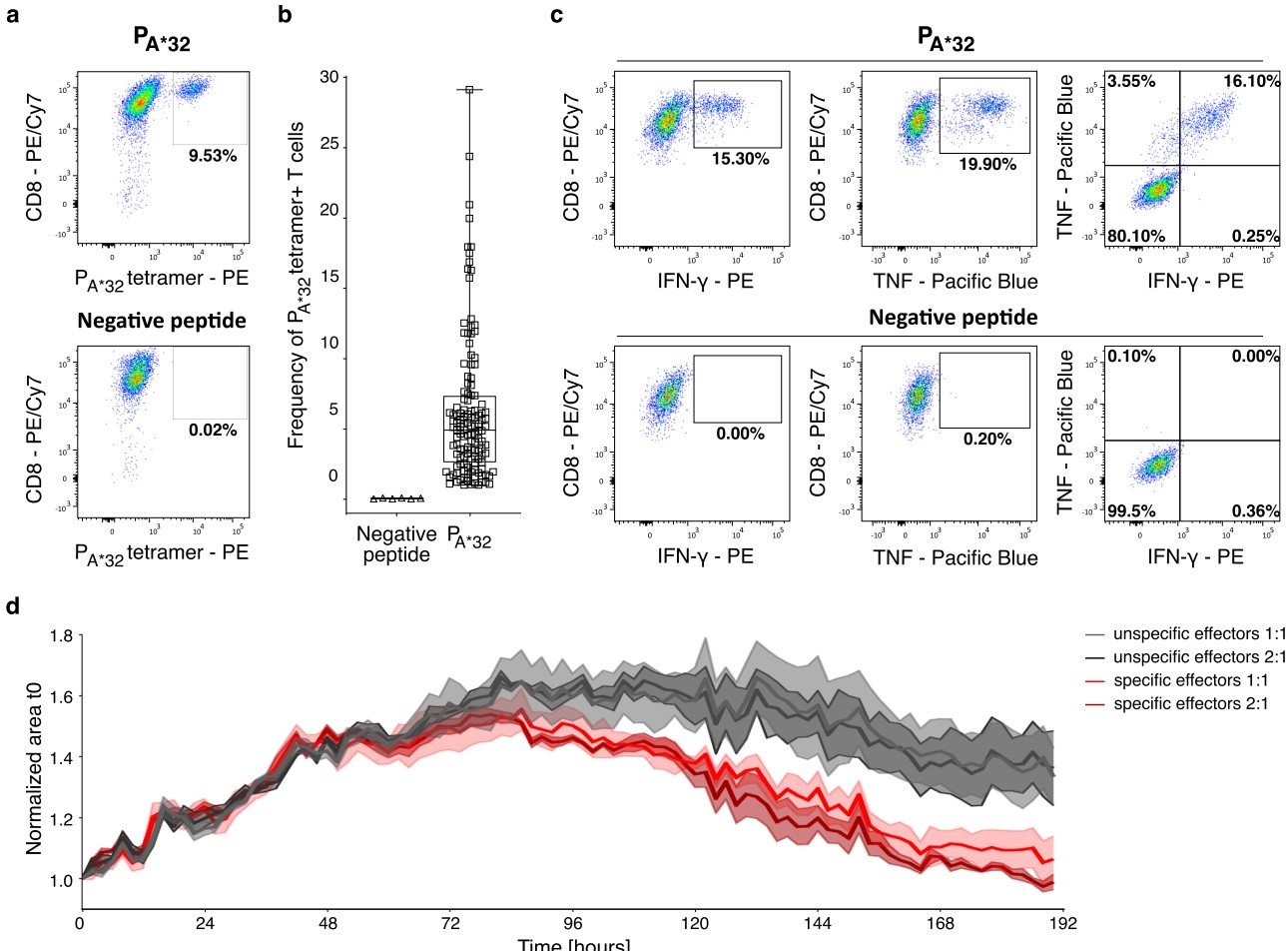

**Fig. 4 | t-neopeptides activate T cells and induce cytotoxic responses.**
**a** Representative example of flow cytometry-based characterization of $P_{A*32}$-specific CD8+ T cells of a healthy volunteer (HV) after in vitro artificial antigen-presenting cell (aAPC)-priming with HLA-A*32-$P_{A*32}$-monomer. **b** Frequencies of $P_{A*32}$-specific CD8+ T cells compared to CD8+ T cells primed with an HLA-matched negative peptide in HVs ($n = 3$, frequency of T cells is indicated per well). All data points are shown, the band indicates the median, and the box indicates the first and third quartiles. **c** Representative example of IFN-γ and TNF production of $P_{A*32}$-specific CD8+ T cells stimulated with $P_{A*32}$ (upper panel) or an HLA-matched negative peptide (lower panel) after aAPC-priming. **d** Specific cell lysis by $P_{A*32}$-specific CD8+ T cells of DAC + SB-treated NCI-H1299 cells at different effector-to-target cell ratios compared to $P_{A*32}$-unspecific CD8+ T cells. Shown is the area of NCI-H1299 cells normalized to time point 0 h (t0) over time until 190 h. Results are shown as mean ± SEM for three independent technical replicates.

However, we did not observe an enrichment of the IFN-α response pathway, as it was previously observed for several ovarian, breast, and colon cancer cell lines upon DNMTi treatment, probably due to a different treatment regimen[14] (Supplementary Fig. 8d–f, bottom). Chiapinelli et al. observed induction of type I interferon response at day 7 or later post Decitabine withdrawal after a single treatment of 72 h with 100 nM Decitabine[14]. We sampled our cells after treating them every day with 500 nM Decitabine for 4 consecutive days. It is possible that we would have observed induction of type I interferon response if we sampled our treated cells at a later time point. We also did not observe increased HLA transcription upon Decitabine treatment as observed by others[28,29] (Supplementary Fig. 8g).

Using a comprehensive reference of predicted novel and non-canonical ORFs, MS-based immunopeptidomics provided direct evidence for the processing and HLA presentation of 45 spectra-validated t-neopeptides induced by DNMT and HDAC inhibition. In line with previous reports[30], a relevant proportion of peptides derived from non-canonical ORFs could not be validated using spectral comparison with synthetic peptides. We could show a clear dependency of sample identification frequency in the biological replicates and the calculated spectral correlation coefficient ($R^2$). This underlines the importance of our approach, which was previously developed for the identification of broadly applicable tumor antigens[5,31,32], to identify treatment-induced

neopeptides (i) by comparative analysis to the DMSO negative control and (ii) by focusing on peptides identified in multiple samples to exclude false discovery rate (FDR)-based wrongly assigned peptides, and thus validates the 15 peptides identified on all three replicates that were exclusively presented in the treated samples as the most promising candidates for further evaluation. This direct proof of naturally presented, treatment-induced novel ORF-derived antigens by immunopeptidomics represents a major advancement in the field, as previously only predictions based on transcriptome analysis or T cell assays were used for the identification of treatment-induced T cell epitopes[18,33]. Whole-cell proteomics identified a total of 4417 known proteins and 272 TINPAT-derived proteins. None of the proteins predicted to give rise to t-neopeptides were identified in the whole-cell proteomics analysis. The AHA SILAC proteomics approach utilized the traditional trypsin/lysC digestion of the proteins. For this reason, an ORF of longer length is required to ensure its identification (Supplementary Fig. 9b). Most of the novel proteins identified arise from longer ORFs (average length = 596 aa) which are predicted to give rise to more tryptic peptides (Supplementary Fig. 9a). On the other hand, most t-neopeptides arise from rather short ORFs (average length = 65 aa) (Supplementary Fig. 9a, c) making their detection in our whole-cell proteomics dataset very difficult. This is in line with the observations made by Ouspenskaia et al.[34].

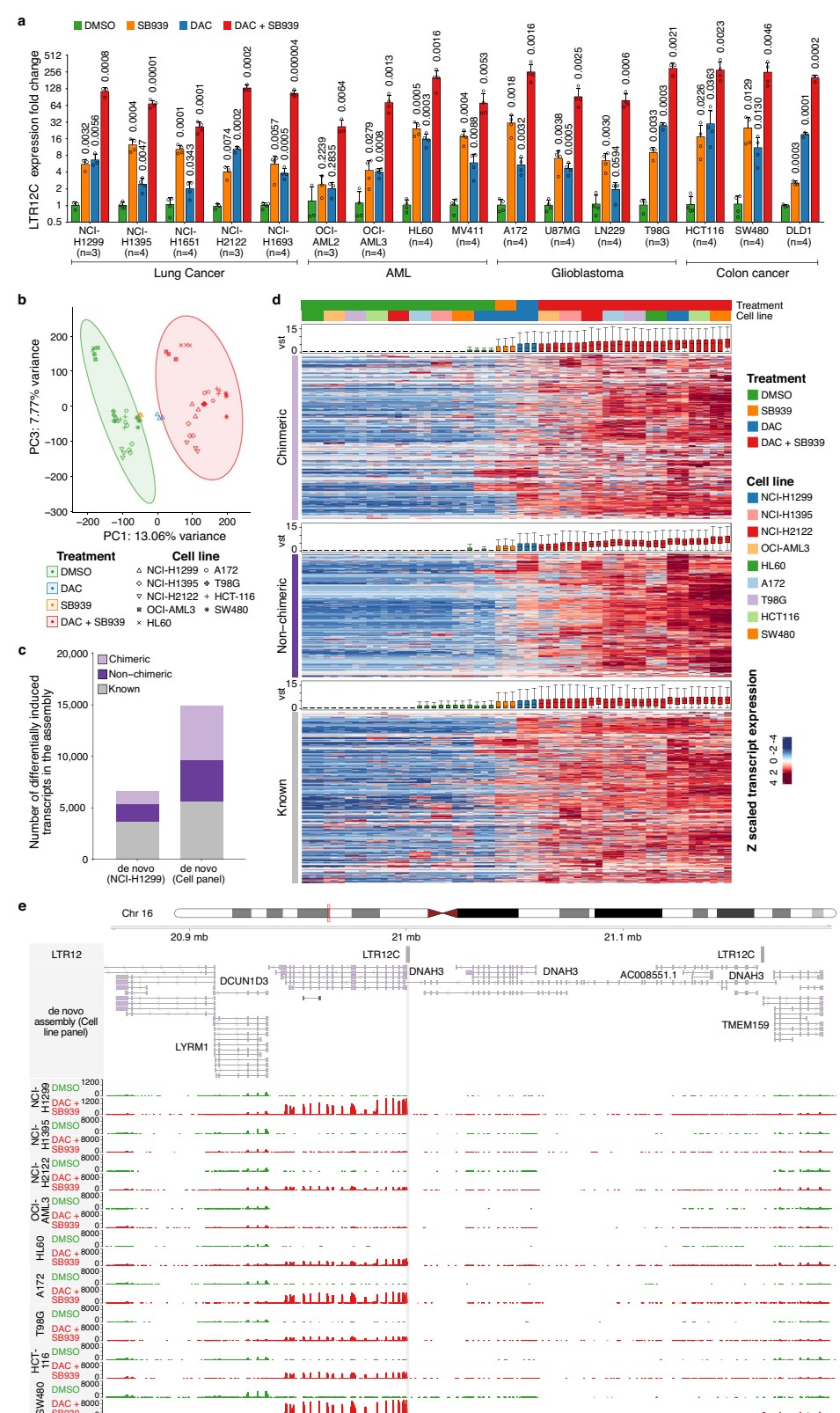

In addition, immunogenicity and cytotoxic T cell activation upon treatment-induced novel ORF-derived HLA ligand presentation were shown. Further analysis of a pan-cancer cell line panel revealed a conserved response of TINPAT induction post-DNMTi and HDACi treatment and allowed us to significantly expand the repertoire of differentially induced transcripts (14,957) and TINPATs (9369). The clinical relevance of t-neopeptides was verified by showing the presence of t-neopeptides in AML patient samples 48/96 h post in vivo treatment with DAC. Together, our findings highlight a mechanism of action by which DNMTi and/or HDACi elicit immune responses and enable the development of immune therapy approaches further exploiting the immunogenicity of t-neopeptides.

In contrast to existing studies that focused on transposable element (TE) -encoded viral neopeptides[15,33,35,36], the use of a de novo

**Fig. 5 | Induction of treatment-induced transcripts is conserved across different cancer entities. a** LTR12C expression analysis performed via quantitative PCR (qPCR) on biological replicates across a panel of cancer cell lines treated with DMSO, SB939, DAC, or DAC + SB939, respectively. Expression is normalized to *GAPDH* and the corresponding DMSO control (mean values ± SD; *n* = the number of biological replicates; *p*-values indicated above each bar, are calculated using a two-tailed unpaired t-test and corrected for multiple comparisons using the Holm-Sidak method). **b** Principal component (PC) analysis of the 5000 most variably expressed transcripts of DAC + SB939 (*n* = 3) and DMSO-treated (*n* = 3) cancer cell lines, assessed via RNA-seq. Transcript expression was quantified using the de novo transcriptome assembly of all cell lines. PC1 and PC3 are visualized. **c** The number of differentially induced transcripts, defined by DESeq2 (adjusted *p*-value < 0.01, log$_2$ fold change >2,) in the DAC + SB939 vs DMSO comparisons of NCI-H1299 cells and the cell line panel de novo transcriptome assembly. **d** Differentially induced transcripts, in the DAC + SB939 vs DMSO comparisons of the cell line panel de novo transcriptome assembly. Colors are only mapped from the 1st to the 99th quantile of the gene expression matrix. The cell line identity was included in the design formula to adjust for cell line-specific differences. Box plots of vst transcript expression are plotted alongside and indicate the largest value within the 1.5 times interquartile range above 75th percentile, 75th percentile, median, 25th percentile, and smallest value within the 1.5 times interquartile range below 25th percentile (*n* = 3 biological replicates). **e** Locus plot of selected chimeric transcript *DNAH3*, identified in the cancer cell line panel post DAC + SB939 treatment, showing (from top to bottom) LTR12 repeats, the de novo cell line panel transcriptome assembly, normalized RNA-seq coverage of DMSO, and DAC + SB939 treated cancer cell lines.

transcriptome assembly, as shown by Attig et al.[37] or Vibert et al.[38], enables the identification of a previously ignored t-neopeptide repertoire. In the future, long-read sequencing technologies might aid in overcoming any potential inaccuracies in the transcriptome assembly of short-read RNA-sequencing data. Further identification of the genomic sequences, encoding for t-neopeptides, revealed that a large proportion of t-neopeptides were not encoded by TE sequences, but rather by downstream exonic sequences. In these cases, TEs only encode for a promoter sequence driving TINPAT expression. Therefore, our pan-cancer de novo transcriptome assembly serves as a blueprint to identify treatment-induced transcripts and t-neopeptides for immunotherapy approaches in malignant diseases treated with DNMTi and HDACi. As most TINPATs also showed a strong induction upon DAC treatment alone, it is likely that DAC treatment could be sufficient to cause the presentation of t-neopeptides, as validated by the identification of t-neopeptides in DAC-treated AML patients in vivo. Previously, Saini et al. demonstrated that HERV-specific T cells are present in AML patients irrespective of DNMTi treatment[33]. In line with these observations, we also detect novel ORF-derived peptides in pre- as well as post-DAC treatment in AML patients. However, the number of novel ORF-derived peptides was largely increased in samples after DAC treatment. Furthermore, we could identify t-neopeptide-specific CD4[+] memory T cells in an AML patient who received in vivo DAC treatment. These findings could explain the observed benefit of DAC therapy in several clinical trials[39–41]. However, this patient also received HDAC inhibitor Valproic acid (VPA) in addition to DAC. Ohtani et al. demonstrated a strong association between LTR12C expression and patient response upon Azacytidine treatment[42]. In contrast, Kazachenka et al. could not find any such association in a larger cohort of patients treated with Azacytidine[43]. Therefore, whether DAC monotherapy can induce t-neopeptide-driven immune response remains unclear and it is likely that patients receiving DAC treatment might benefit from combinatorial therapies targeting t-neopeptides.

Using MS-based immunopeptidomics, our work provides direct evidence for natural cellular processing and HLA-restricted presentation of these t-neopeptides induced by DNMTi and HDACi treatment. This is a necessity for therapeutic targeting of tumor antigens, in particular regarding the distorted correlation between gene expression and HLA-restricted antigen presentation[31,44–46], as well as the limited number of somatic mutations that are ultimately translated, processed, and presented as HLA-restricted neoantigens on tumor cells[44,47–49]. t-neopeptides might represent a novel and superior category of tumor antigens for the design of combinatorial T cell-based immunotherapy approaches comprising vaccines, adoptive T cell transfer, and TCR engineering. This is based on (i) the higher degree of sequence alteration compared to somatic point mutation-derived neoantigens, (ii) increased foreignness compared to treatment-induced CTAs and overexpressed self-antigens[44], and (iii) their applicability independent of the cancer mutational burden. We further provide evidence for in vivo induction of TINPATs, presentation of TINPAT-derived HLA ligands and immunogenicity by validation of pre-existing CD4[+] memory T cells in AML patients treated with DAC. This might explain the positive effect of combinatorial approaches using HMA and immunotherapies comprising DLI and ICI in AML, beyond the increased expression of CTAs[50,51]. Moreover, low-dose application of Decitabine was associated with improved response to ICI treatment in various other cancer entities[52]. Future studies have to evaluate the frequency, tumor/patient specificity, and the safety of t-neopeptides under DAC treatment in large cohorts as well as the clinical relevance of T cells targeting these antigens in vivo.

Together, our work describes a category of immunogenic neoantigens specifically induced by anti-cancer therapies that might serve as prime targets for combinatorial immunotherapeutic approaches in cancer patients.

## Methods
### Patients and blood samples
Peripheral blood mononuclear cells from healthy volunteers as well as AML patients pre-DAC treatment or 48 h and 96 h after the start of DAC treatment were isolated by density gradient centrifugation and stored at −80 °C until further use for subsequent T cell assays or immunopeptidomics analysis. AML patients were intravenously injected with Decitabine (20 mg/m$^2$) each day for five consecutive days. Informed consent was obtained in accordance with the Declaration of Helsinki protocol. The study was performed according to the guidelines of the ethics committee at the medical faculty of the Eberhard-Karls-University and at the University Hospital Tübingen (713/2018B02, 406/2019BO2). Sex was assessed by the clinics and reported in the study. Gender was not recorded in the clinic, due to medical standard procedure and therefore could not be considered. All available patients were included. However due to the small cohort size of two patients a sex- or gender-specific analysis was not feasible. HLA typing was carried out by the University Hospital Tübingen, Germany.

### Cell culture and treatment
NCI-H1299 (ATCC CRL-5803), NCI-H1395 (ATCC CRL-5868), NCI-H1651 (ATCC CRL-5884), NCI-H2122 (ATCC CRL-5985), NCI-H1693 (ATCC CRL-5887), NCI-H1395 (ATCC CRL-5868), OCI-AML2 (DSMZ ACC 99), OCI-AML3 (DSMZ ACC 582), HL60 (ATCC CCL-240), MV411 (DSMZ ACC 102), HCT116 (ATCC CCL-247), SW480 (ATCC CCL-228), and DLD1 (ATCC CCL-221) cells were cultivated in RPMI −1640 supplemented with 10% FCS. A172 (ATCC CCL-1620), U87MG (ATCC HTB-14), LN229 (ATCC CRL-2611), and T98G (ATCC CRL-1690) cells were cultivated in DMEM supplemented with 10% FCS. All cell lines were authenticated and controlled for contamination using the Multiplex cell line authentication and cell contamination test by Multiplexion (Heidelberg, Germany). Cells were treated with 500 nM DAC (A3656, Sigma-Aldrich, Darmstadt, Germany) for 96 h, 500 nM SB939 (Cay10443, Biomol GmbH, Hamburg, Germany) for 18 h, or 500 nM DAC for 78 h and 500 nM SB939 for 18 h. DAC-containing media was refreshed every 24 h. Drugs and doses were selected based on the previous

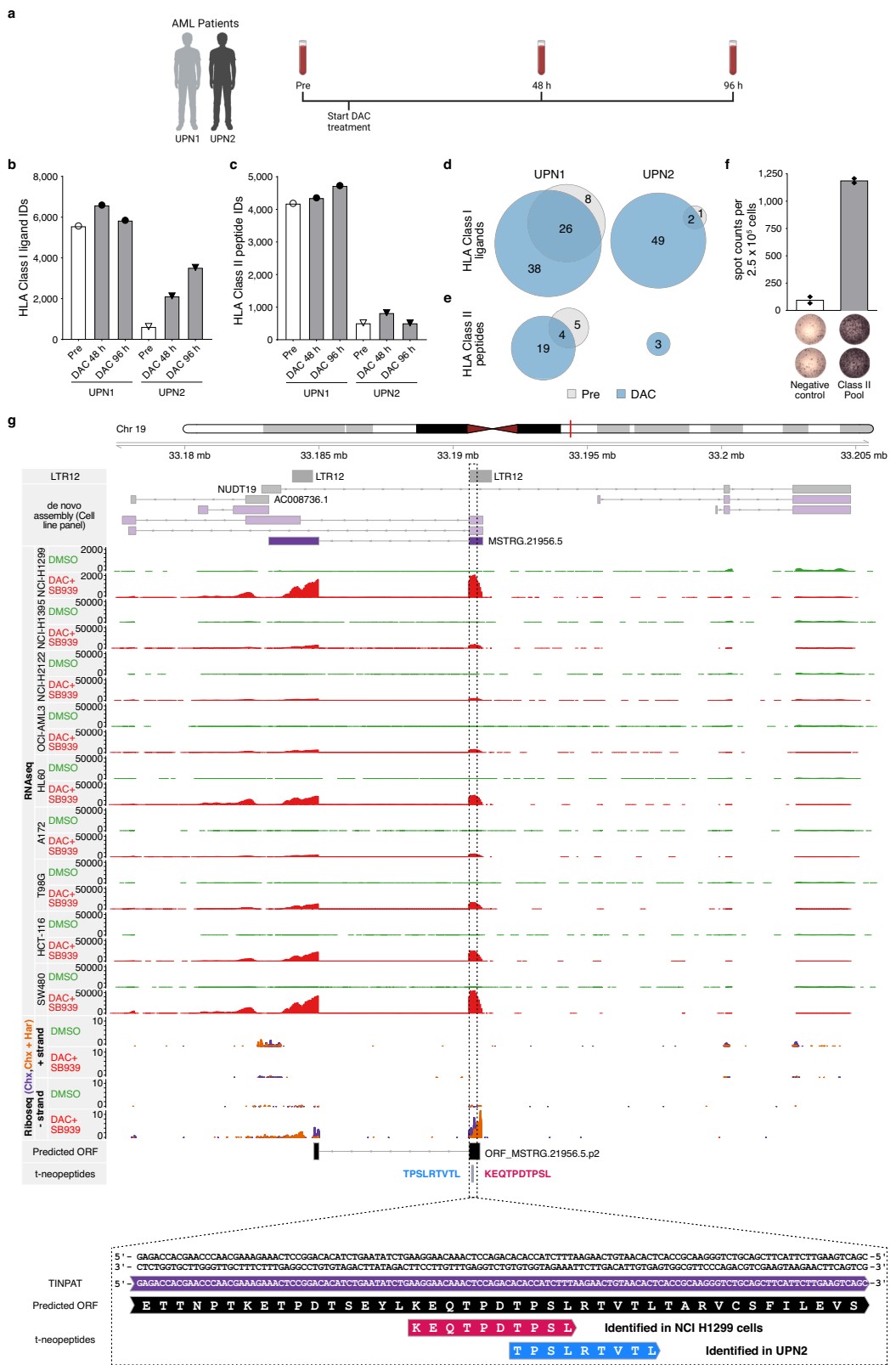

observation that DAC and SB939 induced LTR12C expression in NCI H1299 cells at these doses[18]. For the remaining cell lines, we used the same drugs and doses as NCI H1299.

## RNA-sequencing library preparation

Total RNA from treated cells was extracted using the Qiagen RNeasy Plus Mini kit (Qiagen, Hilden, Germany). RNA integrity was analyzed using High Sensitivity RNA ScreenTape Analysis (Agilent Technologies, Santa Clara, USA). 1 μg RNA (with RNA integrity number > 9) was used for polyA+ RNA sequencing library preparation using TruSeq Stranded mRNA library prep workflow as per the manufacturer's recommendations (Illumina, San Diego, USA). Briefly, polyA+ RNA was purified using oligo-dT magnetic beads and fragmented to 120-210 bp length fragments. Cleaved RNA fragments were reverse transcribed using random

**Fig. 6 | Identification of t-neopeptides in AML patients treated with DAC.**
**a** Schematic overview of the blood sampling from AML patients treated with DAC.
Indicated are the blood collections pretreatment as well as 48 h and 96 h post-DAC
treatment (created with BioRender). **b, c** The number of MS-identified HLA ligands
from AML patients' material. Available samples were from uniform patient number
(UPN) 1 and UPN2, both pre- and post-DAC treatment of (**b**) HLA class I ligands and
(**c**) HLA class II ligands, respectively. **d, e** Overlap analysis of MS-identified novel
ORF-exclusive HLA ligands from AML patients UPN1 and UPN2 after DAC treatment
(48 h and 96 h time point combined) compared to the pretreatment of (**d**) HLA
class I ligands and (**e**), HLA class II ligands. (**f**) t-neopeptide-specific T cell responses
of AML patient UPN9 after DNMTi and valproic acid (VPA) treatment assessed by

IFN-γ ELISPOT assay after in vitro stimulation with the HLA class II peptide pool
compared to the negative control. **g** Locus plot of the non-chimeric transcript
*MSTRG.21956.5* giving rise to a t-neopeptide in NCI-H1299 cells treated with
DAC + SB939 and UPN2 treated with DAC. The locus plot shows (from top to bot-
tom) LTR12 repeats, the de novo cell line panel transcriptome assembly, normal-
ized RNA-seq coverage of DMSO, and DAC + SB939 treated cancer cell lines and
Ribo-seq coverage (from the + and − strand) of DMSO and DAC + SB939 treated
NCI-H1299 cells, as well as the predicted ORF and location of the t-neopeptides
identified in NCI-H1299 cells (red) and UPN2 (blue). The region containing the
t-neopeptides is zoomed in and reversed for improved visualization.

primers followed by second-strand synthesis. The resulting double-
stranded (ds) cDNA fragments were 3′-adenylated and adapter-ligated
using IDT Unique Dual Indexes for Illumina (Integrated DNA Tech-
nologies, Coralville, USA). Adapter ligated ds cDNA fragments were
enriched by PCR for 10-15 cycles and purified using magnetic beads to
yield sequencing-ready libraries. The average length and molarity of
these libraries were determined using the High Sensitivity D1000
ScreenTape assay (Agilent Technologies). 12 to 24 libraries were
pooled (equimolar) to generate a 1 nM multiplexed library which was
sequenced using Nextseq 2000 sequencing system (P3 Reagents, 200
Cycles) in a paired-end sequencing mode (Illumina).

### RNA sequencing data processing

For the analysis of the known reference transcriptome assembly, RNA-
seq data of the NCI-H1299 cell line were processed with the nf-core
RNA-seq pipeline[53] v1.2 with default parameters, unless mentioned
otherwise. In short, raw sequences were aligned to the human refer-
ence genome hg19 by the aligner HISAT2[54] v2.1.0 with the *-reverseS-
tranded* option in *--pairedEnds* mode. Transcripts were assembled
using StringTie[55] v1.3.4d utilizing the GENECODE annotation
v29lift37[56]. Gene counts were generated with the prepDE.py script,
accompanying the StringTie v1.3.4d software.

Coverage tracks were generated from bam files utilizing
deepTools'[57] v3.3.1 function bamCoverage with the option *−normal-
izeUsing RPKM*. Replicates of the different NCI-H1299 treatments were
merged using the software kenUtils (www.github.com/Arunken/
kenutils) v377.

### De novo transcriptome assembly

For the de novo assembly of the NCI-H1299 and entire cell line panel
transcriptomes, sorted bam files from the hg19 alignment were
assembled using StringTie v2.1.1, evoking *-m 200 -f 0.05 -c 1.5 -p 10*
parameters. Individual assemblies of the NCI-H1299 data and the entire
cell line panel were merged with *stringtie --merge* and default para-
meters. De novo assembled transcripts were compared and annotated
with the GENECODE annotation v29lift37 utilizing GFF Utilities[58] v0.11.2
function *gffcompare* with *-R* and *-r* parameters. Transcripts were clas-
sified according to their relationship with the closest reference tran-
script (class_code). The class_code was further simplified by merging
transcripts classified as "s" (intron match on the opposite strand), "x"
(exonic overlap on the opposite strand), "i" (fully contained within a
reference intron), "y" (contains a reference within its intron(s)), "p"
(possible polymerase run-on (no actual overlap)), or "u" (unknown,
intergenic) as non-chimeric (novel) transcripts, transcripts classified as
"m",(retained intron(s), not all introns matched/covered), "j" (multi-
exon with at least one junction match), "e" (single exon transfrag
partially covering an intron, possible pre-mRNA fragment), or "o"
(other same strand overlap with reference exons) as chimeric (novel)
transcripts and transcripts classified as "=" (complete, exact match of
intron chain) as known transcripts. Furthermore, the TSS of transcripts
were annotated to the closest or overlapping TEs using the function
*distanceToNearest* of the R package GenomicRanges[59] v1.38.0.

Quantification of the NCI-H1299 and the cell line panel RNA-seq
data with the individual de novo transcriptome assemblies was per-
formed with the nf-core RNA-seq pipeline v1.2, as described earlier.
Transcripts were assembled using StringTie v1.3.4d and the de novo
transcriptome assemblies. Transcript counts were generated with the
prepDE.py script, accompanying the StringTie v1.3.4d software.

Additionally, we downloaded the human tissue RNA-seq data
from the ENCODE portal[60,61] (https://www.encodeproject.org/) with
the following identifiers: ENCSR725TPW, ENCSR001UXR,
ENCSR612HYR, ENCSR843HXR, ENCSR775KCE, ENCSR629VMZ,
ENCSR635GTY, ENCSR046XHI, ENCSR433XCV, ENCSR071ZMO,
ENCSR693GGB, ENCSR922VBO, ENCSR825GWD, ENCSR686JJB,
ENCSR066FZL, ENCSR102TQN, ENCSR547TNE, ENCSR332MTG,
ENCSR274JRR, ENCSR721HDG, ENCSR995BHD, ENCSR502OTI,
ENCSR917YHC, ENCSR542OHE, ENCSR598KJX, ENCSR880XLM,
ENCSR555BCP, ENCSR146ZKR, ENCSR699YJR, ENCSR675YAS,
ENCSR278UYN, ENCSR510PSL, ENCSR763NOO, ENCSR769LNJ,
ENCSR663IOE, ENCSR618IQY, ENCSR993QGR, ENCSR270OKS,
ENCSR229JRA, ENCSR910QOX, ENCSR719HRO, ENCSR980UEY,
ENCSR741QEH, ENCSR714KDG, ENCSR693CSQ, ENCSR039ICU,
ENCSR085HNI, ENCSR236OON, ENCSR680AAZ, ENCSR129KCJ,
ENCSR448DCX, ENCSR571BML, ENCSR448VSW, ENCSR718CDN,
ENCSR783BUO, ENCSR482VRI, ENCSR817TLH, ENCSR439SPU,
ENCSR999ZCI, and ENCSR396GIH. ENCODE RNA-seq data were pro-
cessed as described earlier and quantified using the de novo assem-
bled NCI-H1299 transcriptome annotation. To compare transcript
expression levels with treated NCI-H1299 cells, TPMs were calculated,
accounting for gene counts and transcript lengths[62].

### Prediction of open reading frames

ORFs were predicted using the software TransDecoder (www.github.
com/TransDecoder/TransDecoder) v2.0. Briefly, transcript sequences
were extracted using the *gffread* function from the software GFF Uti-
lities v0.11.2, and ORFs were predicted using the function *TransDeco-
der.LongOrfs* evoking the parameters *-m 8 -S*. Furthermore, ORFs and
their resulting peptide sequences were filtered for complete or 5′prime
ORFs. For 5′prime ORFs, including multiple Methionines (M), the
longest ORF was selected.

### Differential expression analysis

For the identification of differentially expressed genes in the known
transcriptome annotation, the R package DESeq2[63] v1.26.0 was used.
Gene counts were applied for a group-wise comparison of the different
treatments. An adjusted $p$-value cutoff of <0.05 and an absolute $\log_2$
fold change of >1 were applied to fulfill statistical significance.

Similar analyses were applied for the identification of differen-
tially expressed transcripts in the de novo assembled transcriptome
annotations. In the NCI-H1299 comparison, transcript counts were
applied for a group-wise comparison of the different treatments
(SB939 vs DMSO, DAC vs DMSO, and DAC + SB939 vs DMSO). An
adjusted $p$-value cutoff <0.01 and an absolute $\log_2$ fold change of > 2
was applied to fulfill statistical significance.

To define the treatment effect in the cell line panel, transcript counts were applied to a multi-factor design including the cell line identity and treatment (-*cell_line*+*treatment*), constructed using DESeq2 v1.26.0. The DAC + SB939 vs DMSO specific treatment covariate was extracted using the following contrast: c ("treatment", "DACSB", "DMSO"). Transcripts were considered to be differentially expressed with an adjusted *p*-value cutoff <0.01 and absolute log$_2$ fold change of >2.

### Gene set enrichment analysis

Gene set enrichment analysis (GSEA) of epigenetically silenced genes[64], CTAs[65], and Interferon-alpha response[66] gene sets were performed utilizing the R package clusterProfiler[67] v3.12.0 and the function *GSEA*. Genes were ordered according to their log$_2$ fold change and the fgsea algorithm was applied.

### Analysis of transposable elements' expression

Overlap of aligned reads with TEs was performed using subread's[68] v1.6.4 function *featureCounts* evoking *-p -f -F 'GTF'* parameters. Counts from TE subfamilies were aggregated and normalized using DESeq2 v1.26.0. Hierarchical cluster analysis was applied to the expression of genes/transcripts from the LTR12 subfamily, the *vst* function was applied to normalized counts using the R package DESeq2 v1.26.0, and was visualized using the R package pheatmap[69] v1.0.12.

Differential repeat expression was performed by applying aggregated TE subfamily counts to DESeq2 v1.26.0. To define the treatment effect on repeat expression a multi-factor design, including the cell line identity and treatment (-*cell_line*+*treatment*), was constructed. The DAC + SB939 vs DMSO specific treatment covariate was extracted using the following contrast: c ("treatment", "DACSB", "DMSO").

### Enrichment of transposable elements

The R package LOLA[70] v 1.16.0 was used to enrich TSSs with TE classes, families, and subfamilies. The TE LOLA database was manually generated using the genomic regions of TE classes, families, and subfamilies. The TSS of TINPAs was enriched against a background containing all TSSs from the de novo assembled transcriptome annotation. For the TSSs of TIL encoding transcripts, a background of TSSs of all transcripts that give rise to novel ORFs were used.

### Locus plots

Locus plots were generated using the R package Gviz[71] v1.30.3. CAGE coverages were acquired from GSE81322[18].

### Genomic annotation and homology search of peptide sequences

To define the genomic origin of the identified peptide candidates, the originating ORFs were scanned for the identified peptide sequences and viral peptides[72] using Biostring's[73] v3.42 function *vmatchPattern*. Next, ORF-relative peptide coordinates within the encoding transcripts were mapped to genomic regions by utilizing ensembldb's[74] v2.10.2 function.*to_genome*. For the peptide homology search of t-neopeptides, we have scanned originating ORFs and Uniprot for the identified peptide sequences using Biostring's[74] v3.42 function *vmatchPattern* allowing 0, 1, or 2 mismatches.

### Sequence Polymorphism analysis of the peptide sequences

The 1000 Genomes phase 3 data along with phasing information for the genomic loci corresponding to the peptide sequences was downloaded from the 1000 genome FTP site[75]. We could retrieve the two haplotypes for the 2504 samples. By comparing the sequences of the t-neopeptide genomic regions, % polymorphism at DNA level as well as amino acid level was determined.

### Azidohomoalanine (AHA) pulse SILAC labeling of NCI-H1299 cells for whole-cell proteomics

NCI-H1299 cells were cultured as described above, and DAC was administered for 72 h and SB939 for 18 h for the whole duration of depletion and labeling. The media containing the inhibitors was refreshed every 24 h. For the AHA pulse, SILAC labeling cells were depleted of methionine, lysine, and arginine for 30 min by using RPMI medium 1640 non-GMP formulation (ThermoFisher Scientific, Waltham, USA) devoid of the aforementioned amino acids, supplemented with 10% dialyzed FBS (ThermoFisher Scientific) and 50 U/ml Penicillin−Streptomycin (ThermoFisher Scientific). After depletion, cells were labeled with RPMI medium 1640 non-GMP formulation supplemented with 0.1 mM L-AHA (AnaSpec, Inc), [$^{13}C_6$, $^{15}N_4$] L-arginine, and [$^{13}C$, $^{15}N_2$] ʟ-lysine for heavy isotope labeling or supplemented with 0.1 mM L-AHA), [$^{13}C_6$] L-arginine and [4,4,5,5-D$_4$] L-lysine (Cambridge Isotope Laboratories, Inc., Tewksbury, USA) for intermediate isotope labeling for 4 h. Both labeling media were supplemented with 10% dialyzed FBS and 50 U/ml Penicillin−Streptomycin. Cells were washed three times with PBS and detached by scraping, then centrifuged for 5 min at 1000 × *g*. PBS was discarded and cell pellets were frozen at −80 °C or immediately subjected to protein extraction and enrichment.

### Enrichment of newly synthesized proteins and on-beads digestion

Cell pellets were dissolved in a lysis buffer (0.3 M HEPES, 0.75 M NaCl, 0.1 M CHAPS) containing a 1× protease inhibitor cocktail (Roche, Basel, Switzerland). Cell pellets were sonicated using the following conditions: 10% Amplitude, 50% Duty cycle, 20 cycles, time: 80 s. Cell lysates were centrifuged at 15,000*g* at 4 °C for 30 min and the protein concentration of the supernatants was measured using a BCA protein assay kit (ThermoFisher Scientific). An equal amount of protein extract from the heavy and medium labeled extract was mixed. For the enrichment, the click-IT protein enrichment (ThermoFisher Scientific) was used using the vendor's instructions with minor modifications. Initially, iodoacetamide was added to a final concentration of 55 mM and samples were incubated in the dark for 30 min. The samples were subsequently transferred to tubes containing the alkyne agarose resin. The click-IT reaction takes place in the presence of 200 mM Cu(II)SO$_4$ (Merck, Darmstadt, Germany), 160 mM Cu(I) ligand THPTA (supplied with the kit), 2 M aminoguanidine hydrochloride (Merck), and 2 M sodium ascorbate (supplied with the kit). The reaction mix was incubated for 2 h at 40 °C on a shaking platform while rotating. The beads were incubated with the one-step reduction/alkylation solution, which is the SDS wash buffer (contained in the kit) supplemented with 8 mM Tris(2-carboxyethyl)phosphine (TCEP, Merck) and 33 mM 2-chloroacetamide (CAA, Merck), at 70 °C for 15 min. Samples were washed with SDS wash buffer, water, 5 times with guanidine wash buffer, and 5 times with acetonitrile wash buffer (20% acetonitrile). Resin was resuspended in the digestion buffer (0.1 M Tris pH = 8, 2 mM CaCl$_2$, 5% ACN). Trypsin/LysC was added to the samples in a 50:1 ratio and samples were digested for 18 h. Samples were acidified using formic acid and desalted using Sep-Pak cartridges. Cartridges were washed with neat acetonitrile and then with 60% ACN−0.1% trifluoroacetic acid (TFA, Biosolve, Dieuze, France). The cartridges were equilibrated in 0.1% TFA before sample loading. After sample loading, the cartridges were washed 3 times with 0.1% TFA. Finally, samples were eluted by using 60% ACN−0.1% TFA. Eluate was brought to dryness by Speedvac, and the peptides were resuspended in 0.1% TFA.

### High pH (HpH) reverse-phase peptide fractionation

Peptide samples were subjected to HpH reverse phase fractionation. Samples were basified with the addition of ammonium formate (Merck) to a final concentration of 20 mM. For the fractionation, a Gemini 3 µm C18, 100 × 1 mm, 110 Å (Phenomenex) column was used

that was connected to an Agilent 1260 Infinity Series LC system (Agilent Technologies). The system was operated with a constant flow of 0.1 ml/min with solvent A being 20 mM ammonium formate (pH 10) and solvent B 100% acetonitrile (ACN) ULC/MS grade (Biosolve). The following gradient was applied for the fractionation: 0 to 2 min 0% buffer B, 2 to 60 min the gradient was linearly changed to 65% B, 61 to 62 min the gradient was increased to 85% B, 62 to 67 min the gradient was kept to 85% B, 67 to 85 min 0% B for system re-equilibration. During the gradient application, 40 fractions were collected on a 96 well plate in a vertical manner (A1 to H1 then A2 to H2, etc). Fractions were pooled in a column-wise fashion (i.e. plate column 4 was combined with column plate 3 etc.). The combined fractions were brought to dryness by SpeedVac (ThermoFisher Scientific) and resuspended in an appropriate volume of 0.1% FA. Finally, fractions were cleaned for MS analysis by using an Oasis PRiME HKB µElution plate (Waters) following the instruction of the manufacturer. Cleaned peptide samples were resuspended in an appropriate volume of 0.1% FA and stored at −20 °C until LC-MS/MS analysis.

### LC-MS/MS analysis of NCI-H1299 whole-cell proteomics data

Peptides were separated using an EASY nanoLC 1200 System (ThermoFisher Scientific), equipped with an Acclaim PepMap 100, C18 trapping column (ThermoFisher Scientific), 100 µm × 2 cm, 5 µm, 100 Å, and an Acclaim PepMap RSCL 2 mM C18, 75 mm × 50 cm analytical column (ThermoFisher Scientific). The analytical column was heated to 45 °C by using a MonoSLEEEVE column oven (Analytical Sales and Services). Solvent A was 0.1% FA and solvent B was 80% ACN and 0.1% FA. Sample loading on the trap column took place under the constant pressure of 800 bar for a total volume of 22 µl solvent A. Peptides were eluted from the trapping column with the following gradient at a flow rate of 300 nl/min: 0 to 4 min linear gradient from 3% to 8% B, 4 to 6 min linear gradient 8% to 10% B, 6 to 74 min linear gradient from 10% to 32% B, 74 to 86 min linear gradient from 32% to 50% B, 86 to 87 min gradient increase from 50 to 100% B, 87 to 94 min the gradient was kept to 100% B, 94 to 95 min linear gradient from 100% to 3% B, 95 to 105 min gradient was kept at 3% B.

The analytical column was coupled to a Silica PicoTip Emitter (Scientific Instrument Services) via a zero dead volume connection. Peptides were introduced into the QE-HF mass spectrometer (ThermoFisher Scientific) by using a Nanospray Flex nano spray source (ThermoFisher Scientific) operated at a 2 kV voltage. The ion transfer tube temperature was 275 °C. The following settings were used for data acquisition: Full scan $MS^1$ acquisition was acquired within 350–1500 $m/z$ scan range and 120,000 resolution. The maximum injection time (IT) was set to 32 ms and automatic gain control (AGC) to $3 \times 10^6$ ions. Spectrum data type was set to profile. The top 20 most abundant precursor ions were selected for fragmentation. Ions with unassigned charges and charges of 1 or >5 were excluded. Dynamic exclusion was set to 40 s with a mass tolerance of ±10 ppm. For the $MS^2$ scans, the quadrupole was used with an isolation window of 2.0 $m/z$. For peptide fragmentation higher-energy collisional dissociation (HCD) was used at 26%. $MS^2$ scans were acquired with an AGC target of $1 \times 10^5$ or a maximum IT of 50 ms and a scan range between 200 to 2000 $m/z$. $MS^2$ spectra were acquired in centroid mode.

### Analysis of the raw MS data from NCI-H1299 whole-cell proteomics analysis

The acquired raw files were analyzed by using different databases:

**Raw data analysis using TINPAT produced database.** The raw data were searched using Maxquant version 2.0.3.0 against the RNA-seq TINPAT database. The multiplicity option was set to 3 representing the SILAC labels, for medium labels Arg6 and Lys4 were selected whereas for heavy labels Arg10 and Lys8 were chosen. Enzyme digestion was set to Trypsin allowing for a maximum of up to 2 missed cleavages.

Oxidation (M), Acetyl (Protein N-term), and deamidation (NQ) were set as variable modifications and carbamidomethyl (C) as fixed. The minimum unique peptides for the protein identification option were set to 1 while peptide and protein hits were filtered at FDR of 1% with a minimum peptide length of 7 amino acids. The reversed sequences of the aforementioned target database were used as a decoy database for FDR calculation. The second peptide search option was enabled. The match between runs option was enabled with a matching time window of 0.4 min and the alignment time window set to 20 min. Depended peptides option was deactivated. Label min.ratio count option was set to 0 and unique + razor peptides were considered for quantification. Peptides that were modified by Oxidation (M), acetylation (Protein N-term), and deamidation (NQ) were considered for SILAC quantification with unmodified counterparts being discarded. Intensity-based absolute quantification (iBAQ) values were also enabled. All other remaining Maxquant options were left to their default settings.

**Raw data analysis using the human canonical database.** The raw data were searched using Maxquant version 1.6.8.0 against the human canonical and reviewed database (downloaded November 2018). The multiplicity option was set to 3 representing the SILAC labels, for medium labels Arg6 and Lys4 were selected whereas for heavy labels Arg10 and Lys8 were chosen. Enzyme digestion was set to Trypsin allowing for a maximum of up to 2 missed cleavages. Oxidation (M), Acetyl (Protein N-term), and deamidation (NQ) were set as variable modifications and carbamidomethyl (C) as fixed. The minimum unique peptides for protein identification was set to 2 while peptide and protein hits were filtered at a FDR of 1% with a minimum peptide length of 7 amino acids. The reversed sequences of the human canonical database were used as a decoy database for FDR calculation. The second peptide search option was enabled. The match between runs option was enabled with a matching time window of 0.4 min and the alignment time window set to 20 min. The depended peptides option was deactivated. iBAQ calculation was also enabled. All other Maxquant options were left to their default settings.

### Bioinformatic analysis of Maxquant output data from NCI-H1299 whole-cell proteomics analysis

Maxquant output tables were analyzed with Microsoft Excel and R statistical software environment version 4.0.3. More specifically quality control of the acquired raw data was assessed with the use of the PTXQC proteomics package. The Maxquant proteinGroups.txt output table was used for the analysis of both the differentially translated TINPATs and differentially translated proteins upon SB939 and DAC administration. The table was filtered for the contaminants, reverse, and only identified by site entries. Reported SILAC ratios were $\log_2$ transformed and the normalized SILAC ratios for the H/M were chosen for the downstream differential analysis. The table was then filtered for entries reporting at least 4 valid values in the 7 biological replicates. For the TINPAT dataset strongly translated candidates were included in the analysis. These candidates are defined as entries that have an intensity value in at least 4 out 7 replicates for either the heavy or the medium channel whereas there is no intensity reported in the other channel (i.e. entries with at least 4 out of 7 replicates in the heavy channel and no intensity value reported in the medium channel). In order for these candidates to be represented quantitatively, the following imputation method was applied. For the biological replicates where an intensity value was reported in the respective SILAC channel but a missing value was present in the other, a random intensity value was sampled from a left-censored normal distribution. The distribution had a mean, calculated from the minimum intensity value for that specific biological replicate, multiplied by a factor of 0.75 to ensure non-detectability and a standard deviation calculated from the intensities for that biological replicate. Non-normalized ratios were created for those 35 entries which were then median. Finally, those ratios were

incorporated into the table with the other ratios. Before applying differential analysis using Limma, the table was imputed by using random forest imputation (MissForest, R package). Limma differential analysis was assessed by plotting the non-adjusted *p*-values to assess the distribution. For the table derived for the protein analysis, entries reporting a ratio in 4 out of 7 biological replicates were kept. Limma analysis was performed without data imputation and non-adjusted *p*-values were assessed for Limma performance. ggplot2 was used for the creation of the volcano plots.

## Quantification of HLA surface expression

HLA surface expression of the NCI-H1299 cell line was analyzed using the QIFIKIT bead-based quantitative flow cytometric assay according to the manufacturer's instructions as described before (Agilent Technologies)[76]. In brief, samples were stained with the pan-HLA class I-specific monoclonal antibody (mAb) W6/32 (produced in-house), the HLA-DR-specific mAb L243 (produced in-house), or IgG isotype control (400202, BioLegend, San Diego, USA). Flow cytometric analysis was performed on a FACSCanto II Analyzer (BD Biosciences, New Jersey, USA).

## Isolation of HLA ligands

HLA class I and HLA class II molecules were isolated by standard immunoaffinity purification[77] using the pan-HLA class I-specific mAb W6/32, the pan-HLA class II-specific mAb Tü 39, and the HLA-DR-specific mAb L243 (all produced in-house) to extract HLA ligands.

## Analysis of HLA ligands by liquid chromatography-coupled tandem mass spectrometry (LC-MS/MS)

Peptide samples were separated by reversed-phase liquid chromatography (nanoUHPLC, UltiMate 3000 RSLCnano, ThermoFisher Scientific) and subsequently analyzed in an on-line coupled Orbitrap Fusion Lumos mass spectrometer (ThermoFisher Scientific). Samples were analyzed in five technical replicates. Sample volumes of 5 μl with shares of 17% were injected onto a 75 μm × 2 cm trapping column (ThermoFisher Scientific) at 4 μl/min for 5.75 min. Peptide separation was subsequently performed at 50 °C and a flow rate of 300 nL/min on a 50 μm × 25 cm separation column (PepMap C18, ThermoFisher Scientific) applying a gradient ranging from 2.4 to 32.0% of ACN over the course of 90 min. Eluting HLA class I peptides were ionized by nanospray ionization and analyzed in the mass spectrometer implementing a top speed (3 s) CID (collision-induced dissociation) method generating fragment spectra with a resolution of 30,000, a mass range limited to 400−650 *m/z*, and positive charge states 2−3 selected for fragmentation. HLA class II peptides were analyzed with a HCD (Higher-energy C-trap dissociation) method, a mass range limited to 400−1000 *m/z*, and positive charge states 2−5 selected for fragmentation.

## Immunopeptidomics data processing

Data processing was performed as described previously[78]. The Proteome Discoverer (v1.4, Thermo Fisher) was used to integrate the search results of the SequestHT search engine (University of Washington[79]) against the human proteome (Swiss-Prot database, 20,279 reviewed protein sequences, August 21st, 2019) accompanied by the complete ORF sequences. Precursor mass tolerance was set to 5 ppm and fragment mass tolerance was set to 0.02 Da. For the immunopeptidome analysis the post-translational modification (PTM) oxidized methionine was allowed as a dynamic modification. The FDR (estimated by the Percolator algorithm 2.04[80]) was limited to 5% for HLA class I and 1% for HLA class II. Peptides with an XCorr below 1 were excluded from the analysis. HLA class I annotation was performed using SYFPEITHI 1.0[81] and NetMHCpan 4.1[82]. For the detailed PTM search analysis oxidation of methionine, tryptophan, and histidine, methylation of aspartic acid, histidine, isoleucine, leucine, lysine, and arginine, acetylation of lysine, carbamidomethylation of cysteine, and

phosphorylation of serine, threonine, and tyrosine were allowed as dynamic modifications.

## Spectrum validation

Spectrum validation of the experimentally eluted peptides was performed by computing the similarity of the spectra with corresponding synthetic peptides measured in a complex matrix. The spectral correlation was calculated between the MS/MS spectra of the eluted and the synthetic peptide as described previously[83]. For the high-throughput spectral validation of all identified TINPAT-derived peptides a visualization tool based on nf-core/mhcquant (v2.4.0) to retrieve ion annotations of the experimentally and synthetically obtained fragmentation spectra was used.

## Refolding

Biotinylated HLA:peptide complexes were manufactured as described previously[84] and tetramerized using PE-conjugated streptavidin (ThermoFisher Scientific) at a 4:1 molar ratio.

## Induction of peptide-specific CD8⁺ T cells with aAPCs

Priming of peptide-specific cytotoxic T lymphocytes was conducted using antigen-presenting cells (aAPC). In detail, 800,000 streptavidin-coated microspheres (Bangs Laboratories, Fishers, USA) were loaded with 200 ng biotinylated HLA:peptide monomer and 600 ng biotinylated anti-human CD28 monoclonal antibody (clone 9.3, in-house production). CD8⁺ T cells were isolated from PBMCs of healthy donors using a MACS kit (Miltenyi Biotec, 130-045-201) and cultured with 4.8 U/μl IL-2 (R&D Systems, Minneapolis, USA) and 1.25 ng/ml IL-7 (PromoCell GmbH, Heidelberg, Germany) in round bottom 96-well plates. Weekly stimulation with aAPCs (200,000 aAPCs per 1 × 10⁶ CD8⁺ T cells) and 5 ng/ml IL-12 (PromoCell GmbH) was performed for four cycles. IL-2 (4.8 U/μl) was added three days after each stimulation cycle. The CD8⁺ T cells were left to rest for one week prior to the flow cytometry-based tetramer staining analysis[85].

## Cytokine, surface marker, and tetramer staining

The functionality of peptide-specific T cells was analyzed by surface marker and intracellular cytokine staining (ICS) as described previously[86,87]. Cells were pulsed with 10 μg/ml of respective peptide and incubated with 10 μg/ml Brefeldin A (Merck) and 10 μg/ml Golgi-Stop (BD Biosciences) for 12–16 h. Staining was performed using Cytofix/Cytoperm (BD Biosciences), Aqua live/dead (1:400 dilution, ThermoFisher Scientific), PE/Cy7 anti-human CD8 (1:400 dilution, Cat# 737661, RRID: AB_1575980, Beckman Coulter, Brea, USA), Pacific Blue anti-human TNF (1:120 dilution, Cat# 502920, RRID: AB_528965, BioLegend), and PE anti-human IFN-γ mAB (1:200 dilution, Cat# 506507, RRID: AB_315440, BioLegend). PMA and ionomycin (Merck) served as a positive control. Negative control peptides with matching HLA restrictions were used: KYPENFFLL for HLA-A*24 (source protein: PP1G_HUMAN) and RIKQIINMW for HLA-A*32 (source protein: gp160_HIV-1). Gating strategies applied for the analyses of flow cytometry-acquired data are provided in Supplementary Fig. 8a.

The frequency of peptide-specific CD8⁺ T cells after aAPC-based priming was determined by Aqua live/dead (1:400 dilution, ThermoFisher Scientific), PE/Cy7 anti-human CD8 (1:400 dilution, Cat# 737661, RRID: AB_1575980, Beckman Coulter) and HLA:peptide tetramer-PE staining. Cells of the same donor primed with an irrelevant control peptide (for A*24 TYSEKTTLF (source protein: MUC16_HUMAN), for A*32 RIKQIINMW (source protein: gp160_HIV-1), for A*03 ALHQPLVHR (source protein: MK01_HUMAN), and for B*07 RPKENVTIM (source protein: LY9_HUMAN)) and stained with the tetramer containing the test peptide were used as a negative control. Priming was considered successful if the frequency of peptide-specific CD8⁺ T cells was ≥ 0.1% of CD8⁺ T cells within the viable single-cell population and at least three-fold higher than the frequency of peptide-specific CD8⁺ T cells in

the negative control. The same evaluation criteria were applied for ICS results. Samples were analyzed on a FACS Canto II cytometer (BD Biosciences). The gating strategy applied for tetramer staining analysis of flow cytometry-acquired data is provided in Supplementary Fig. 8b.

### Amplification of peptide-specific T cells and interferon-γ (IFN-γ enzyme-linked immunospot (ELISpot) assay

PBMCs from AML patients after DNMTi therapy were pulsed with 5 μg/ml per HLA class II peptide and the HLA class II negative control peptide ETVITVDTKAAGKGK (source protein: FLNA_HUMAN) was also used for stimulation. Cells were cultured for 12 days adding 20 U/ml IL-2 (Novartis, Basel, Switzerland) on days 2, 5, and 7. Peptide-stimulated PBMCs were analyzed by IFN-γ ELISpot assay on day 12 with anti-IFN-γ antibody (clone 1-D1K, 2 μg/mL, MabTech), anti-IFN-γ biotinylated detection antibody (clone 7 B6 1, 0.3 μg/mL, MabTech), ExtrAvidin-Alkaline Phosphatase (1:1000 dilution, Sigma-Aldrich) and BCIP/NBT (5 bromo 4-chloro 3 indolyl-phosphate/nitro-blue tetrazolium chloride, Sigma-Aldrich)[78]. Spots were counted using an ImmunoSpot S6 analyzer (CTL, Cleveland, OH, USA) and T cell responses were considered positive when >10 spots/500 000 cells were counted, and the mean spot count was at least three-fold higher than the mean spot count of the negative control.

### Cytotoxicity assays

Cytotoxicity analyses were performed using the IncuCyte S3 Live-Cell Analysis System (Sartorius, Göttingen, Germany). NCI-H1299 cells constitutively expressing the mCherry fluorescent protein were treated for 4 days with DAC + SB939 prior to the cytotoxicity assay and then transferred to a 96-well plate (2000 cells/well). NCI-H1299 cells were co-cultured with peptide-specific CD8+ T cells generated by aAPC-based priming, or unspecific CD8+ T cells as control, in a 2:1 or 1:1 effector to target cell ratio, respectively. Live-cell imaging pictures were taken every 2 h with ×10 magnification. To quantify NCI-H1299 cells, the red fluorescence areas were normalized to the respective measurement at t = 0 h. Error bars represent the standard error of the mean.

### RT-qPCR expression analysis

RT-qPCR expression analysis for LTR12C was performed as described previously[18]. Briefly, 1 μg RNA was reverse transcribed using random hexamers and Superscript III Reverse Transcriptase (ThermoFisher Scientific) according to the manufacturer's instructions. qPCR was performed on the Roche Lightcycler 480 system using primaQUANT CYBR mix (Steinbrenner Laborsysteme GmbH, Wiesenbach, Germany) and primers against LTR12C (TCACTCTTTGGGTCCACACT and TGGAGTTGTTCGTTCCTCCC) or GAPDH (AGCCACATCGCTCAGACAC and GCCCAATACGACCAAATCC). Quantification was performed using the ΔΔCt method and the housekeeping gene GAPDH was used for normalization.

### Riboseq library preparation

DMSO and DAC + SB939-treated NCI-H1299 cells were further treated with 100 μg/ml Cycloheximide (CHX, C4859, Sigma-Aldrich) alone, or in combination with 2 μg/ml Harringtonine (HAR, LKT-H0169, Biomol GmbH) and incubated at 37 °C for 10 min. Ribosome profiling was performed as described previously[88].

### Riboseq data processing

Sample adapters were trimmed using cutadapt (v2.6) and demultiplexed with barcode_splitter from FASTX-toolkit (v0.0.6). Small fragments (x < 30 nt) were dropped. Unique molecular identifier (UMI) extraction was performed using umi_tools (v1.0.1). By BLAST-Like Alignment Tool, rRNA reads were filtered and discarded. The rRNA index for Rn5s, Rn18s, Rs5-8s1 was constructed from NCBI annotations.

The remaining reads were aligned with Spliced Transcripts Alignment to a Reference (STAR) (v2.7.10) to GRCh37/hg19. Following, PCR duplicates were removed using umi_tools.

### Software and statistical analysis

All figures and statistical analyses were generated using R v3.6[89] and GraphPad Prism (GraphPad Software, San Diego). The cytotoxicity analyses were evaluated with a two-way analysis of variance (ANOVA). Schematic overview figures were created with BioRender.com.

### Reporting summary

Further information on research design is available in the Nature Portfolio Reporting Summary linked to this article.

## Data availability

The Ribo-seq and RNA sequencing data generated in this study have been deposited in the NCBI Gene Expression Omnibus (GEO) database under the primary accession number GSE209777. The whole cell proteomics mass spectrometry data generated in this study have been deposited in the ProteomeXchange Consortium (http://proteomecentral.proteomexchange.org) via the PRIDE[54] partner repository under the dataset identifier PXD035748. The immuno-peptidomics mass spectrometry data generated in this study have been deposited to the ProteomeXchange Consortium via the PRIDE partner repository under the dataset identifier PXD035750. The H1299 CAGE sequencing publicly available data used in this study are available in the GEO database under accession codes GSE81322[18]. The human tissue RNA sequencing publicly available data used in this study are available in the ENCODE database under accession codes: ENCSR725TPW, ENCSR001UXR, ENCSR612HYR, ENCSR843HXR, ENCSR775KCE, ENCSR629VMZ, ENCSR635GTY, ENCSR046XHI, ENCSR433XCV, ENCSR071ZMO, ENCSR693GGB, ENCSR922VBO, ENCSR825GWD, ENCSR686JJB, ENCSR066FZL, ENCSR102TQN, ENCSR547TNE, ENCSR332MTG, ENCSR274JRR, ENCSR721HDG, ENCSR995BHD, ENCSR502OTI, ENCSR917YHC, ENCSR542OHE, ENCSR598KJX, ENCSR880XLM, ENCSR555BCP, ENCSR146ZKR, ENCSR699YJR, ENCSR675YAS, ENCSR278UYN, ENCSR510PSL, ENCSR763NOO, ENCSR769LNJ, ENCSR663IOE, ENCSR618IQY, ENCSR993QGR, ENCSR270OKS, ENCSR229JRA, ENCSR910QOX, ENCSR719HRO, ENCSR980UEY, ENCSR741QEH, ENCSR714KDG, ENCSR693CSQ, ENCSR039ICU, ENCSR085HNI, ENCSR236OON, ENCSR680AAZ, ENCSR129KCJ, ENCSR448DCX, ENCSR571BML, ENCSR448VSW, ENCSR718CDN, ENCSR783BUO, ENCSR482VRI, ENCSR817TLH, ENCSR439SPU, ENCSR999ZCI, and ENCSR396GIH[60,61]. The remaining data are available within the Article, Supplementary Information or Source Data file. Source data are provided with this paper.

## Code availability

Data analysis is based on publicly available software listed in the "Methods" section. Custom code for the analysis of next-generation sequencing data was deposited at https://github.com/HeyLifeHD/TINAT.

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

## Acknowledgements
The authors would like to thank the Genomics and Proteomics Core Facility (GPCF, DKFZ, Heidelberg), Omics IT and Data Management Core Facility (ODCF, DKFZ, Heidelberg), and the Single Cell Open Lab (DKFZ, Heidelberg). Work was supported by funding from the Deutsche Forschungsgemeinschaft, (DFG FOR2674 subprojects A01 to C.P.; A09 to C.P., M.L., and A.G.). The work was supported in part by the German-Israeli Helmholtz International Research School Cancer-TRAX (HIRS-0003 to C.P. and J.H.), the German Center for Lung Diseases (DZL) to C.P. and J.H. and German Consortium for Translational Cancer Research (DKTK) to C.P., H.-G.R., and J.S.W., as well as, the Deutsche Forschungsgemeinschaft under Germany's Excellence Strategy (Grant EXC2180-390900677) to H.-G.R. and J.S.W., the Deutsche Forschungsgemeinschaft (DFG, German Research Foundation, Grant WA 4608/1-2) to J.S.W., the Wilhelm Sander Stiftung (Grant 2016.177.3) to J.S.W., the José Carreras Leukämie-Stiftung (Grant DJCLS 05R/2017) to J.S.W., the Zentren für Personalisierte Medizin (ZPM) to J.S.W., the Deutsche Krebshilfe (German Cancer Aid, 70114948) to J.S.W., and the Bundesministerium für Bildung und Forschung (BMBF, FKZ:01KI20130) to J.S.W.. D.P. has been supported by the SFB1324 "Mechanisms and Functions of WNT Signalling" of the German Research Foundation (DFG).

## Author contributions
A.G., J.B., J.H., J.S.W., and C.P. designed the study. H.-G.R. provided feedback on the study design. A.G. and O.M. performed drug treatments of NCI-H1299 cells and the cancer cell line panel. J.B. and M.Du. performed and analyzed immunopeptidome experiments, J.B., B.K., and M.M. conducted in vitro T cell experiments. J.H. processed and conducted analyses of next-generation sequencing data, including de novo transcriptome assemblies. D.P., D.S., M.D., and J.K. performed whole-cell proteomics. F.L.P., A.G., and E.S. performed Riboseq. M.R., J.S.W., M.L., and T.M. conducted patient data and sample collection as well as medical evaluation and analysis. Y.L. performed the sequence polymorphism analysis. J.B. and J.S. carried out the spectra validation of all the identified peptides. A.G., J.B., and J.H. created the figures. A.G., J.B., J.H., J.S.W., and C.P. drafted the manuscript.

## Funding

## Competing interests
The University Hospital Tübingen and German Cancer Research Center (DKFZ, Heidelberg) is in the process of applying for a patent application (Patent number: EP23170469.3, Title: Neoepitope immunogenic peptides for cancer treatment and diagnosis) covering t-neopeptides that lists A.G., J.B., J.H., J.S.W., and C.P. as inventors. The other authors declare no competing interests.

## Additional information

[1]Cancer Epigenomics, German Cancer Research Center (DKFZ), Heidelberg, Germany. [2]Department of Peptide-based Immunotherapy, University of Tübingen and University Hospital Tübingen, Tübingen, Germany. [3]Institute for Cell Biology, Department of Immunology, University of Tübingen, Tübingen, Germany. [4]Cluster of Excellence iFIT (EXC2180) "Image-Guided and Functionally Instructed Tumor Therapies", University of Tübingen, Tübingen, Germany. [5]German-Israeli Helmholtz Research School in Cancer Biology, Heidelberg, Germany. [6]German Center for Lung Research, (DZL) partner site Heidelberg, Heidelberg, Germany. [7]Division of Proteomics of Stem Cells and Cancer, German Cancer Research Center (DKFZ), Heidelberg, Germany. [8]Heidelberg University, Medical Faculty, Heidelberg, Germany. [9]Translational Control and Metabolism, German Cancer Research Center (DKFZ), Heidelberg, Germany. [10]Quantitative Biology Center (QBiC), University of Tübingen, Tübingen, Germany. [11]Clinical Collaboration Unit Translational Immunology, German Cancer Consortium (DKTK), Department of Internal Medicine, University Hospital Tübingen, Tübingen, Germany. [12]Department of Hematology, Oncology and Stem Cell Transplantation, University Medical Center Freiburg, Faculty of Medicine, University of Freiburg, Freiburg, Germany. [13]German Cancer Consortium (DKTK), Heidelberg, Germany. [14]Present address: Department of Hematology and Central Hematology Laboratory, Inselspital, Bern, University Hospital, University of Bern, Bern, Switzerland. [15]These authors contributed equally: Ashish Goyal, Jens Bauer, Joschka Hey. [16]These authors jointly supervised this work: Juliane S. Walz, Christoph Plass. ✉e-mail: juliane.walz@med.uni-tuebingen.de; c.plass@dkfz.de

