## [Peer Review File · Nature Communications]

DNMT and HDAC inhibition induces immunogenic neoantigens from human endogenous retroviral element-derived transcriptsReviewers' Comments:

Reviewer #1:

Remarks to the Author:

Goyal and colleagues investigated the presentation and potential antigenicity of TINPATs-derived peptides, produced by cancer cell lines, and perhaps AML (although the information about AML is unclear in the text; see below), upon treatment with DAC and HDACi. The work expands the previous publication from the same group (doi:10.1038/ng.3889), wherein they demonstrated that DNMT and HDAC inhibitors induce cryptic transcription start sites encoded in long terminal repeats.

My main comments of the study are:

- the authors applied an improved strategy for the transcript detection, followed by SILAC proteomics, which lead to the identification of 272 proteins from the predicted ORFs. Can the authors show the ratio between TINPATs and the TINPATs-derived proteins, as compared to the ratio between canonical transcripts and cognate proteins? I am curious to understand if the likelihood of translation + identification of TINPATs-derived proteins is similar to the canonical proteins.

- The HLA-I immunopeptidomics using a tailored database containing the theoretical TINPATs derived proteins led to the identification of 112 peptides exclusively assigned to the novel ORF repertoire. 63 were treatment specific. These peptides were not derived from the 272 proteins (see above). I think more analysis should be done in that direction to explain this discrepancy.

- The Supplem Data of the immunopeptidomics should provide all needed information for QC and comparison of results obtained with other strategies. For example, scan number, RT, score, etc for each peptide should be shown in Supplem Data 7, 8, etc. These info might be available on the PRIDE projects, but I was not provided with the access info to control that. It is very important that all needed information to re-use the dataset are available without the need to request them to the authors.

- I would like to see some further validation of the 112 peptides mentioned above rather than comparison to 15 synthetic peptides. They latter represent the peptides present in all technical replicates and thus, likely, the most convincing PSMs. What about all others? If the authors are not keen of comparing all 112 with synthetic peptides, at least they could perform a comparison with the predicted MS2 spectra by ProSIT. The spectral angle of these non canonical peptides should be compared to that of the canonical. I am curious to see the outcome of this analysis. Chong et al (<https://doi.org/10.1038/s41467-020-14968-9>) compared their non canonical peptides to heavy-labeled synthetic peptides and many were not confirmed.

- in the M & M, although the proteomics analysis is well described, the immunopeptidomics section is scarce in terms of information, which are needed to evaluate the manuscript. For example, did the author do an open search for PTMs of canonical peptides to exclude that the 112 peptides are not PTM-modified canonical peptide sequences? This type of analysis has been done to invalidate the identification of post-translationally spliced peptides in other studies (e.g. doi: 10.1002/pmic.202000176). Can it be that the DAC and HDACi modify the PTM repertoire and thus lead to mis assignment of sequences? This consideration might be connected by the authors to the fact that none of the 112 peptides in immunopeptidomes derived from the 212 proteins derived from TINPATs.

- I would not use the term 'neoepitopes' to define the 112 peptides in the immunopeptidomes. What are the evidences that they are epitopes? The authors proved that naive CD8 T cells specific for the noncanonical peptides can be primed from healthy donors, but this does not mean that they are immunogenic in cancer and upon treatment.

- Regarding the potential immunogenicity of the TINPATs-derived peptides suggested through the

study of 2 AML patients, I did not understand from what cells the 124 ORF-derived peptides were eluted? Full PBMCs? AML cells? It should be AML cells, no?

Also, no antigenicity assays have been done in this section. Are these peptides recognised only by AML patient upon treatment? This information would provide a significant upgrade of the study, and it should be carried out on a larger cohort.

Reviewer #2:

Remarks to the Author:

Noteworthy results

The authors expand on their previous publication on treatment-induced novel-polyadenylated transcripts (Brock et al. Nature Genetics). Here, they use in silico predictions, immunopeptidomics, and AML patient- material after decitabine treatment, to characterise the potential of (mainly) LTR12-derived non-reference transcripts for immunogenic HLA-I epitopes. Importantly, the authors provide strong evidence for at least 15 HLA-I neo-epitopes upon DAC/SB393 treatment for cells with AML-origins.

Overall comment on publication

I expect the study will be of immediate interest to research into the consequences of decitabine / 5-azacitidine administration in AML, MDS and CML and physicians that consider combination and sequential treatment regimens with immunotherapy agents. It is also of interest to the growing field of long terminal repeats as alternative promoters.

The manuscript is well written and presented. It is warranted to address a limited number of concerns in the primary data analysis to support the conclusions, as listed below

Reviewer's Concerns

Below a number of concerns on phrases and presentation of the data which should be addressed before publication.

In the context of immunotherapy, we need to ask of neo- or tumour-antigens:

- are they individualised or shared between patients
- is immunological tolerance and expression in normal tissue expected or not
- are they expressed and presented by a sufficiently large amount of the tumour cells (or by tumour stem cells, proliferating cells, etc.)

This is in particular relevant for neo-antigens discovered via peptidomics that are (partially) derived from genomics repeats

To this end, I would expect the authors to address the following:

(1a) peptide homology of t-neoepitopes

- Homology with other hypothetical peptides: a summary of a blast search (or similar) against a transcriptome reference (+ all TINATs) or the human genome; highlighting the potential of other loci to generate the neo-epitope in question (+/- 1 amino acid difference).

Context: The authors partially address this concern in Figure 3 and state that 25 out of 38 t-neoepitopes are derived from a single transcript and outside of the LTR12 sequences. However, the epitope in Figure 6f arises from the LTR12 sequence, and their example in Figure 3b shows an epitope

mapped to a presumed unique locus extremely close to the annotated LTR12 element. Since the boundaries of retroelements in mammalian genomes are (i) not 100% precisely annotated and (ii) recombination events leading to stand alone LTR elements often involve incomplete matching of the repeat sequences and recombination with other repeat sequences, it seems very likely to me that some of these t-neoepitopes still derive from sequences with close homology across multiple loci in the human genome; which is an important piece of information in this study

(1b)

- Homology between individuals: a summary of the polymorphism in the sequence of the genomic loci generating the amino acids of the t-neoepitopes shown in Table1 in the 1000 genome projects (or similar resource)

(2) Rephrase abstract

The abstract is attempting to draw attention by using big numbers a bit too freely.

-> e.g. 3,023 ERV-derived, treatment-induced novel polyadenylated transcripts (TINPATs), encoding for 61,426 open reading frames

The respective figure 2 also shows ~72,000 ORFs from known and annotated transcripts. In principle, the authors would be able to derive a false discovery rate for the 72,000 known ORFs to be truly translated, by asking how many have accumulated evidence for being translated across the scientific knowledgebase using a source of their choice (proteomics, ribosome sequencing, etc). I would pose that less than 10% of the 72,000 ORFs of known transcripts will have evidence of producing peptides. The likelihood of chimeric/non-chimeric transcripts without purifying selection pressure will be lower than that. I would find it more informative if the authors lead in the abstract with the number of spectra-validated peptides rather than the hypothetical number of ORFs.

In a similar direction, the final statement in the abstract aims to high:

Our findings highlight a novel mechanism of ERV-derived neoantigens in epigenetic and immune therapies

ERV-derived neoantigens have been hypothesized and demonstrated for decades, and regarding the potential for DNMT inhibitors to induce them, the author's own prior publication in Brock et al. states in the abstract:

The resulting transcripts frequently splice into protein-coding exons and encode truncated or chimeric ORFs translated into products with predicted abnormal or immunogenic functions.

(3) missing item in discussion

The identification of >15 HLA-I tumour antigens is a major advance in the field.

As part of the discussion, the authors lay out the idea of using such knowledge and the presence/diversity/abundance of t-neo-epitopes for engineering clinical response. They need to weigh the available evidence in favour but also the available evidence contradicting this idea. I found at least one study that did not find association with response for ERV expression levels after 5-azacytidine in MDS patients: Kazachenka Genome Medicine volume 11, Article number: 86 (2019) [link].

I suspect there are more of them across tumour indications.

(4) Expand and correct Figure 1/ Supp Figure 2e.

For the comparison of TINPAT expression upon treatment versus normal tissue, the expression of the chimeric TINPATs should be compared against the major annotated isoform(s) of the gene the TINPAT splices into. If for ~3000 TINPATs, >2100 are not at all expressed in a particular tissue, the boxplot in

Figure S1e will average closely to 0 (as shown). If the authors would plot 2000 tissue-specific genes in the same manner, we would falsely conclude that all those genes are lowly expressed in human. This is harder to control for in non-chimeric novel transcripts, hence I suggest to present the two classes separately

Miscellaneous comments

sentence on line137: non-chimeric TINPATs were predicted to have low protein-coding potential ... could be explained by the smaller average length of non-chimeric TINPATs

The authors continue on line 151ff showing all possible ORFs from chimeric and non-chimeric TINPATs and Figure 2 a/b/c illustrate nicely that non-chimeric TINPATs indeed have comparably lower protein-coding potential (smaller ORFs and less of them). The sentence at line137 is superfluous and somewhat confusing.

Reviewer #3:

Remarks to the Author:

The manuscript "DNMT and HDAC inhibition induces immunogenic neoantigens from human endogenous retroviral element-derived transcripts" from Plass and colleagues is a thorough exploration of the proteins and peptides induced by epigenetic therapy (DNMTi and HDACi) in cancer cell lines. They go beyond characterization of the proteome to immunopeptidomics and show that novel peptides are presented on MHC complexes of cancer cell lines after epigenetic therapy. Interestingly, the presented peptides were distinct from the full length proteins they identified from proteomics. The use of a large panel of cell lines from different cancer types is commendable and the final experiments showing that in vivo DAC treatment can induce presentation of immunogenic peptides on HLA in AML patients show that this presentation is happening in vivo.

I have the following questions:

1. Doses of Dac and SB939 should be discussed for Figure 1. In the Methods they state that the Dac + SB939 treatment was 78h Dac, 18h SB939 but the Dac alone was 96 h. How were these drugs and these doses selected? How do they relate to the drugs and doses used for patients in the clinic?
2. In Figure 1, they mention type I interferon responses were not induced because of a different treatment regimen; how does theirs differ? This is a surprise as they do observe a strong induction of ERV transcripts.
3. The immunogenicity experiments in Figure 4 are very encouraging and show that the novel peptides can be immunogenic. However more information needs to be added about the protocol they used to stimulate the CD8 T cells to achieve this response.
4. The novel peptides seem to be induced at similar levels by DAC and DAC/SB939 and the data in Figure 6 is only from DAC-treated patients. The authors should comment on the specificity of these LTR-derived peptides and whether DNMTi treatment alone is sufficient to generate an immune response. In addition, the dose of DAC and treatment regimen in the patients should be specified.
5. Others have reported MHC upregulation at the transcript and protein level by DNMTi treatment, which would improve antigen presentation to T cells. Do the authors observe this in their treated cell lines?

6. Figure 6 shows nicely that some of the LTR-derived peptides are presented on MHC of AML patients; this seems to occur both with and without DAC. Do these presented peptides activate T cells from the AML patients? This would prove an immune response *in vivo*.

Reviewer #1, expertise in immunopeptidomics

(Remarks to the Author):

Goyal and colleagues investigated the presentation and potential antigenicity of TINPATs-derived peptides, produced by cancer cell lines, and perhaps AML (although the information about AML is unclear in the text; see below), upon treatment with DAC and HDACi. The work expands the previous publication from the same group (doi:10.1038/ng.3889), wherein they demonstrated that DNMT and HDAC inhibitors induce cryptic transcription start sites encoded in long terminal repeats.

Authors' response: We thank the reviewer for the constructive criticism of our study. Please find below a detailed point-by-point response addressing the raised questions and concerns.

My main comments of the study are:

Reviewer comment 1:

The authors applied an improved strategy for the transcript detection, followed by SILAC proteomics, which lead to the identification of 272 proteins from the predicted ORFs. Can the authors show the ratio between TINPATs and the TINPATs-derived proteins, as compared to the ratio between canonical transcripts and cognate proteins? I am curious to understand if the likelihood of translation + identification of TINPATs-derived proteins is similar to the canonical proteins.

Authors' response: De novo assembly identified 207,411 known transcripts and 30,695 novel transcripts (line 101). Differential expression analysis identified 3627 known transcripts and 3,023 TINPATs (line 116). Whole-cell proteomics identified a total of 4417 known proteins and 272 TINPAT-derived proteins (line 156, line 299).

TINPAT-derived proteins	272	Canonical proteins	4417
TINPATs	3023	Canonical transcripts	207,411
Ratio	0.089	Ratio	0.021

Changes to the manuscript:

Since these numbers are already available in the manuscript, we did not make any additional changes to the manuscript. However, we performed additional analysis and addressed this issue in more detail (Please refer to Reviewer comment 2).

Reviewer comment 2:

The HLA-I immunopeptidomics using a tailored database containing the theoretical TINPATs derived proteins led to the identification of 112 peptides exclusively assigned to the novel ORF repertoire. 63 were treatment specific. These peptides were not derived from the 272 proteins (see above). I think more analysis should be done in that direction to explain this discrepancy.

Authors' response: We thank the reviewer for making this point. The AHA SILAC-based newly synthesized proteome approach utilized the traditional trypsin/lysC digestion of the proteins. This makes it orthogonal to the immunopeptidomics workflow as it is designed for identifying tryptic peptides. In addition, the AHA labeling approach renders the identification of the N-terminal protein part rather difficult as it is retained covalently attached to the beads. For these reasons, an ORF of longer length is required to ensure proper identification. To illustrate this we plotted ORF length against the number of tryptic peptides from each predicted ORF and highlighted the whole cell proteomics identified TINPAT-derived proteins (in red) as well as the predicted t-neopeptide harboring ORFs (in blue). TINPAT-derived proteins arise from longer ORFs (average length=596 aa), predicted to give rise to more tryptic

peptides. On the other hand, t-neopeptides arise from rather short ORFs (Average length = 65 aa) making their detection in our whole-cell proteomics dataset very difficult.

Changes to the manuscript:

This analysis is now included as Supplementary figure 9a,b in the revised manuscript and the relevant text have also been revised (lines 299-308). Previously designated Figure 3c has now been moved to Supplementary Figure 9c to improve the flow of the manuscript.

Reviewer comment 3:

The Supplem Data of the immunopeptidomics should provide all needed information for QC and comparison of results obtained with other strategies. For example, scan number, RT, score, etc for each peptide should be shown in Supplem Data 7, 8, etc. These info might be available on the PRIDE projects, but I was not provided with the access info to control that. It is very important that all needed information to re-use the dataset are available without the need to request it from the authors.

Authors' response: We apologize for the missing information regarding the quality control of the presented immunopeptidomics data.

Changes to the manuscript:

We modified the relevant Supplementary Data files by adding additional information. Furthermore, we provided the reviewer account information for the corresponding PRIDE project in the materials and methods section of the revised manuscript (line 820).

Reviewer comment 4:

I would like to see some further validation of the 112 peptides mentioned above rather than comparison to 15 synthetic peptides. They latter represent the peptides present in all technical replicates and thus, likely, the most convincing PSMs. What about all others? If the authors are not keen of comparing all 112 with synthetic peptides, at least they could perform a comparison with the predicted MS2 spectra by Prosit. The spectral angle of these non canonical peptides should be compared to that of the canonical. I am curious to see the outcome of this analysis. Chong et al (<https://doi.org/10.1038/s41467-020-14968-9>) compared their non canonical peptides to heavy-labeled synthetic peptides and many were not confirmed.

Authors' response: We thank the reviewer for this valuable comment to improve the quality of the presented data. We analyzed the complete list of 112 identified peptides by comparing their MS-identified spectra using synthetic peptides. 70 peptides could be validated whereas 42 failed the validation. Of note, we could show a clear dependency of sample identification frequency in the biological replicates and the calculated spectral correlation coefficient (R^2) from the synthetic peptide validation in comparison to the comparative profiling analysis. This underlines the importance of our approach to identifying true treatment-induced neopeptides by comparative analysis to the DMSO negative control and focusing on peptides identified in multiple samples to exclude FDR-based wrongly assigned peptides. This further validates the 15 peptides identified on all three replicates that were exclusively presented in the treated samples as the most promising candidates for further evaluation.

Changes to the manuscript:

The new spectra validation data is included in a new Figure 2i, k, Supplementary Fig. 3e and Supplementary Data 7. The relevant text has been updated accordingly (lines 175-181).

Reviewer comment 5:

In the M & M, although the proteomics analysis is well described, the immunopeptidomics section is scarce in terms of information, which are needed to evaluate the manuscript. For example, did the author do an open search for PTMs

of canonical peptides to exclude that the 112 peptides are not PTM-modified canonical peptide sequences? This type of analysis has been done to invalidate the identification of post-translationally spliced peptides in other studies (e.g. doi: 10.1002/pmic.202000176). Can it be that the DAC and HDACi modify the PTM repertoire and thus lead to mis assignment of sequences? This consideration might be connected by the authors to the fact that none of the 112 peptides in immunopeptidomes derived from the 212 proteins derived from TINPATs.

Authors' response: We thank the reviewer for this valuable suggestion to improve the quality of our manuscript. We modified the materials and method section of the revised manuscript to indicate more clearly the parameters applied for the immunopeptidomics analysis.

In addition, we conducted a second immunopeptidomics data analysis including the most abundant PTMs (oxidation of methionine, tryptophan, and histidine, methylation of aspartic acid, histidine, isoleucine, leucine, lysine, and arginine, acetylation of lysine, carbamidomethylation of cysteine, and phosphorylation of serine, threonine, and tyrosine) in the search. In this PTM search 44 of the 63 treatment-exclusive peptides (70%), 16 of the 29 shared peptides (55%), and 7 of the 20 DMSO control exclusive peptides (35%) were identified with an overall lower amount of peptide identifications due to the increased search space. This is in line with the data from the synthetic peptide validation described in more detail in reviewer comment 4. No differential presentation of the entirety of PTM-peptides could be observed between the immunopeptidome of DAC and HDACi-treated cells and the DMSO treatment cells. Moreover, no increase of lysine-acetylation was observed in the immunopeptidome of the DAC and HDACi-treated cells.

The missing overlap between the 112 peptides identified in the immunopeptidome analysis and the 212 proteins derived from TINPATs identified in the proteomic analysis is addressed in reviewer comment 2.

Changes to the manuscript:

We modified the materials and method section of the revised manuscript to indicate more clearly the parameters applied for the immunopeptidomics analysis (line 707).

The novel data on PTM-analysis was included in a novel Supplementary Fig. 3f and described in the result section of the revised manuscript (line 181).

Reviewer comment 6:

I would not use the term 'neoepitopes' to define the 112 peptides in the immunopeptidomes. What are the evidences that they are epitopes? The authors proved that naive CD8 T cells specific for the noncanonical peptides can be primed from healthy donors, but this does not mean that they are immunogenic in cancer and upon treatment.

Authors' response and changes to the manuscript: We have renamed t-neoepitopes in the revised manuscript as t-neopeptides.

Reviewer comment 7:

Regarding the potential immunogenicity of the TINPATs-derived peptides suggested through the study of 2 AML patients, I did not understand from what cells the 124 ORF-derived peptides were eluted? Full PBMCs? AML cells? It should be AML cells, no?

Also, no antigenicity assays have been done in this section. Are these peptides recognised only by AML patient upon treatment? This information would provide a significant upgrade of the study, and it should be carried out on a larger cohort.

Authors' response: We thank the reviewer for this important comment to improve the significance of our findings. The immunopeptidomics analysis of the two AML patients was performed from patients PBMCs as both patients had AML blast counts of 53% (UPN1) and 30% (UPN2) in the peripheral blood.

Immunogenicity testing of the TINPAT-derived peptides identified in the treated AML patients was not possible in the respective patients, as patient material was very scarce and there was no possibility for further sample collection. However, we agree with the reviewer that immunogenicity testing of these antigens would provide a significant upgrade of our findings. Therefore, we selected nine HLA class II-restricted peptides identified in the immunopeptidomics analysis of the AML patients and tested these in Interferon- γ (IFN- γ) Enzyme-Linked Immunospot (ELISpot) assays using PBMCs of eight AML patients that had undergone DNMTi treatment to test for peptide-specific CD4⁺ memory T cells. We could show antigen-specific CD4⁺ memory T cells in one out of the eight DNMTi-treated AML patients.

In addition, we analyzed three HLA class I t-neopeptides identified in the two AML patients for their ability to induce antigen-specific T cells using artificial antigen-presenting cell (aAPC)-based primings of CD8⁺ T cells of healthy volunteers (HV). We could show the induction of antigen-specific T cells against the A*03:01 ligand RTDSSLLEK.

Changes to the manuscript:

We now included new data showing the immunogenicity of t-neopeptides in the revised manuscript Figure 6f and Supplementary Figure 7a (lines 252-259), carried out on a larger cohort of AML patients and healthy volunteers. Additionally, we modified the materials and methods section describing the immunopeptidomics analysis of the two AML patients in more detail (line 370-371).

We thank the reviewer for their valuable suggestions and hope that they are content with the newly added experiments, data, and information included in the revised manuscript and find the manuscript now acceptable for publication in Nature Communications.

Reviewer #2, expertise in ERV-derived transcripts and bioinformatics

(Remarks to the Author):

Noteworthy results

The authors expand on their previous publication on treatment-induced novel-polyadenylated transcripts (Brock et al. Nature Genetics). Here, they use in silico predictions, immunopeptidomics, and AML patient- material after decitabine treatment, to characterise the potential of (mainly) LTR12-derived non-reference transcripts for immunogenic HLA-I epitopes. Importantly, the authors provide strong evidence for at least 15 HLA-I neo-epitopes upon DAC/SB393 treatment for cells with AML-origins.

Overall comment on publication

I expect the study will be of immediate interest to research into the consequences of decitabine / 5-azacitidine administration in AML, MDS and CML and physicians that consider combination and sequential treatment regimens with immunotherapy agents. It is also of interest to the growing field of long terminal repeats as alternative promoters.

The manuscript is well written and presented. It is warranted to address a limited number of concerns in the primary data analysis to support the conclusions, as listed below

Authors' response: We thank the reviewer for the positive feedback on our study. Please find below a detailed point-to-point response addressing the raised questions and concerns.

Reviewer's Concerns

Below a number of concerns on phrases and presentation of the data which should be addressed before publication.

Reviewer comment 1a:

In the context of immunotherapy, we need to ask of neo- or tumour-antigens:

- are they individualised or shared between patients
- is immunological tolerance and expression in normal tissue expected or not
- are they expressed and presented by a sufficiently large amount of the tumour cells (or by tumour stem cells, proliferating cells, etc.)

This is in particular relevant for neo-antigens discovered via peptidomics that are (partially) derived from genomics repeats

To this end, I would expect the authors to address the following:

(1a) peptide homology of t-neoepitopes

- Homology with other hypothetical peptides: a summary of a blast search (or similar) against a transcriptome reference (+ all TINATs) or the human genome; highlighting the potential of other loci to generate the neo-epitope in question (+/- 1 amino acid difference).

Context: The authors partially address this concern in Figure 3 and state that 25 out of 38 t-neoepitopes are derived from a single transcript and outside of the LTR12 sequences. However, the epitope in Figure 6f arises from the LTR12 sequence, and their example in Figure 3b shows an epitope mapped to a presumed unique locus extremely close to the annotated LTR12 element. Since the boundaries of retroelements in mammalian genomes are (i) not 100% precisely annotated and (ii) recombination events leading to stand alone LTR elements often involve incomplete matching of the repeat sequences and recombination with other repeat sequences, it seems very likely to me that some of these t-neoepitopes still derive from sequences with close homology across multiple loci in the human genome; which is an important piece of information in this study.

Authors' response: We thank the reviewer for their valid criticism. To address this issue, we took the 38 unique peptides (found in at least 2 of the 3 biological replicates) and investigated their occurrence in Uniprot as well as the predicted DAC+SB939 induced ORFs (from Figure 2a), allowing an increasing number of mismatches (0, 1, or 2). At 0 mismatches allowed, 25/38 (65%) t-neopeptides were uniquely mapped. After allowing for 1 mismatch, 23/38 (60%) of t-neopeptide still remained uniquely mapped. 16/38 (42%) t-neopeptides were uniquely mapped even after allowing for 2 mismatches indicating that these peptides are sufficiently divergent from known or TINPAT-derived proteins.

Changes to the manuscript:

This data is now included in the manuscript as Supplementary Figure 4c,d and Supplementary data 8. The manuscript text has been updated accordingly (lines 189-194)

Reviewer comment 1b:

(1b) Homology between individuals: a summary of the polymorphism in the sequence of the genomic loci generating the amino acids of the t-neopeptides shown in Table1 in the 1000 genome projects (or similar resource)

Authors' response: We thank the reviewer for this suggestion and have now performed an analysis of polymorphism at DNA as well as peptide level for the 15 t-neopeptides mentioned in Table 1. Overall, we found very low polymorphism at the loci encoding these peptides.

Changes to the manuscript:

The results are summarized in Supplementary data 9. 12/15 (80%) t-neopeptides had less than 0.001 % polymorphism. The relevant text in results (lines 202-205) and methods (lines 515-520) sections have been updated accordingly.

Reviewer comment 2

(2) Rephrase abstract

The abstract is attempting to draw attention by using big numbers a bit too freely.

-> e.g. 3,023 ERV-derived, treatment-induced novel polyadenylated transcripts (TINPATs), encoding for 61,426 open reading frames

The respective figure 2 also shows ~72,000 ORFs from known and annotated transcripts. In principle, the authors would be able to derive a false discovery rate for the 72,000 known ORFs to be truly translated, by asking how many have accumulated evidence for being translated across the scientific knowledgebase using a source of their choice (proteomics, ribosome sequencing, etc). I would pose that less than 10% of the 72,000 ORFs of known transcripts will have evidence of producing peptides. The likelihood of chimeric/non-chimeric transcripts without purifying selection pressure will be lower than that. I would find it more informative if the authors lead in the abstract with the number of spectra-validated peptides rather than the hypothetical number of ORFs.

In a similar direction, the final statement in the abstract aims to high:

Our findings highlight a novel mechanism of ERV-derived neoantigens in epigenetic and immune therapies

ERV-derived neoantigens have been hypothesized and demonstrated for decades, and regarding the potential for DNMT inhibitors to induce them, the author's own prior publication in Brock et al. states in the abstract:

The resulting transcripts frequently splice into protein-coding exons and encode truncated or chimeric ORFs translated into products with predicted abnormal or immunogenic functions.

Authors' response and changes to the manuscript: We have now modified the abstract to reflect the suggested changes by the reviewer (lines 33-45).

Reviewer comment 3:

(3) missing item in discussion

The identification of >15 HLA-I tumour antigens is a major advance in the field.

Authors' response and changes to the manuscript: We agree with the reviewer on this point and have thus highlighted this in the discussion section of the revised manuscript (lines 295-299).

Reviewer comment 4:

As part of the discussion, the authors lay out the idea of using such knowledge and the presence/diversity/abundance of t-neo-epitopes for engineering clinical response. They need to weigh the available evidence in favour but also the available evidence contradicting this idea. I found at least one study that did not find association with response for ERV expression levels after 5-azacytidine in MDS patients: Kazachenka Genome Medicine volume 11, Article number: 86 (2019) [link]. I suspect there are more of them across tumour indications.

Authors' response: We thank the reviewer for making this point. Ohtani et al. (Cancer research, 2020) demonstrated a strong association between LTR12C expression and patient response upon Azacytidine treatment. However, Kazachenka et al. (Genome Medicine, 2019) could not find such association in a larger cohort of patients treated with Azacytidine. Therefore, whether DAC monotherapy can induce t-neopeptide driven immune response remains unclear and it is likely that patients receiving DAC treatment might benefit from combinatorial therapies targeting t-neopeptides.

Changes to the manuscript: We have adapted the discussion section accordingly in the revised manuscript (lines 337-342).

Reviewer comment 5:

(4) Expand and correct Figure 1/ Supp Figure 2e.

For the comparison of TINPAT expression upon treatment versus normal tissue, the expression of the chimeric TINPATs should be compared against the major annotated isoform(s) of the gene the TINPAT splices into. If for ~3000 TINPATs, >2100 are not at all expressed in a particular tissue, the boxplot in Figure S1e will average closely to 0 (as shown). If the authors would plot 2000 tissue-specific genes in the same manner, we would falsely conclude that all those genes are lowly expressed in human. This is harder to control for in non-chimeric novel transcripts, hence I suggest to present the two classes separately

Authors' response:

We thank the reviewer for pointing out this flaw in the analysis. Besides TINPATs, now we also plot the TPMs for transcripts that were annotated to the same gene locus as TINPATs but were classified as known (labeled as known isoforms).

Changes to the manuscript:

Supplementary Figure 2e has been updated accordingly.

Reviewer comment 6:

Miscellaneous comments

sentence on line137: non-chimeric TINPATs were predicted to have low protein-coding potential ... could be explained by the smaller average length of non-chimeric TINPATs

The authors continue on line 151ff showing all possible ORFs from chimeric and non-chimeric TINPATs and Figure 2 a/b/c illustrate nicely that non-chimeric TINPATs indeed have comparably lower protein-coding potential (smaller ORFs and less of them). The sentence at line137 is superfluous and somewhat confusing.

Authors' response and changes to the manuscript: We have now removed the following line from the main text: "However, this could be explained by the smaller average length of non-chimeric TINPATs, which is considered in the protein-coding potential calculation."

We thank the reviewer for their valuable suggestions and hope that they are content with the newly added experiments, data, and information included in the revised manuscript and find the manuscript now acceptable for publication in Nature Communications.

Reviewer #3, expertise in the combination of epigenetic therapies with immunotherapies

(Remarks to the Author):

The manuscript “DNMT and HDAC inhibition induces immunogenic neoantigens from human endogenous retroviral element-derived transcripts” from Plass and colleagues is a thorough exploration of the proteins and peptides induced by epigenetic therapy (DNMTi and HDACi) in cancer cell lines. They go beyond characterization of the proteome to immunopeptidomics and show that novel peptides are presented on MHC complexes of cancer cell lines after epigenetic therapy. Interestingly, the presented peptides were distinct from the full length proteins they identified from proteomics. The use of a large panel of cell lines from different cancer types is commendable and the final experiments showing that in vivo DAC treatment can induce presentation of immunogenic peptides on HLA in AML patients show that this presentation is happening in vivo.

Authors’ response: We thank the reviewer for their kind words and extremely positive feedback on our study. Please find below a detailed point-to-point response addressing the raised questions and concerns.

I have the following questions:

Reviewer comment 1:

1. Doses of Dac and SB939 should be discussed for Figure 1. In the Methods they state that the Dac + SB939 treatment was 78h Dac, 18h SB939 but the Dac alone was 96 h. How were these drugs and these doses selected? How do they relate to the drugs and doses used for patients in the clinic?

Authors’ response: Drugs and doses were selected based on our previous observation that DAC and SB939 induced LTR12C expression in NCI H1299 cells at these doses (Brocks et al, Nature genetics, 2017). For the remaining cell lines, we used the same drugs and doses as NCI H1299. We cannot rule out that for some of these cell lines, the used doses are sub-optimal. The AML patients treated in the clinics were intravenously injected with Decitabine (20 mg/m²) each day for five days in a row, and PBMCs for the immunopeptidomics analysis were isolated before the start of the treatment as well as 48 h and 96 h post-treatment.

Changes to the manuscript:

We have now updated the methods section to include these details (lines 385-387 and lines 371-372).

Reviewer comment 2:

2. In Figure 1, they mention type I interferon responses were not induced because of a different treatment regimen; how does theirs differ? This is a surprise as they do observe a strong induction of ERV transcripts.

Authors’ response: Chiapinelli et al (Cell, 2015), observed induction of type I interferon response at day 7 or later post Decitabine withdrawal after a single treatment of 72 hours with 100nM Decitabine. We extracted RNA from our cells after treating them with 500 nM Decitabine for 96 h (Decitabine-containing media was refreshed every 24 h). It is possible that we would have observed induction of type I interferon response if we extracted RNA at a later time point.

Changes to the manuscript:

We have now added this information to the discussion section (lines 286-291). Furthermore, to improve the flow of the manuscript, we now moved the previously designated Supplementary Figure 1d-i to Supplementary Figure 8a-f. The relevant text has also been moved to the discussion section (lines 281-286).

Reviewer comment 3:

3. The immunogenicity experiments in Figure 4 are very encouraging and show that the novel peptides can be immunogenic. However, more information needs to be added about the protocol they used to stimulate the CD8 T cells to achieve this response.

Authors' response and changes to the manuscript: We thank the reviewer for this positive feedback and valuable suggestion to improve the quality of our manuscript. We modified the materials and method section of the revised manuscript to describe more clearly the artificial antigen-presenting cell (aAPC)-based priming protocol used to stimulate the CD8⁺ T cells (lines 728-738).

Reviewer comment 4:

4. The novel peptides seem to be induced at similar levels by DAC and DAC/SB939 and the data in Figure 6 is only from DAC-treated patients. The authors should comment on the specificity of these LTR-derived peptides and whether DNMTi treatment alone is sufficient to generate an immune response. In addition, the dose of DAC and treatment regimen in the patients should be specified.

Authors' response: We thank the reviewer for making this point. As suggested, we now include a comment in the discussion section stating that it is likely that DAC treatment alone could be sufficient to cause the presentation of t-neopeptides as validated by the identification of t-neopeptides in DAC-treated AML patients in vivo. Furthermore, we could identify peptide-specific CD4⁺ memory T cells targeting a panel of HLA class II presented t-neopeptides in one out of the eight AML patients who received in vivo DAC treatment. However, this patient also received HDAC inhibitor Valproic acid (VPA) in addition to DAC. Therefore, whether DAC monotherapy can induce t-neopeptide driven immune response remains unclear.

The AML patients treated in the clinics were intravenously injected with Decitabine (20 mg/m²) each day for five days in a row, and PBMCs for the immunopeptidomics analysis were isolated before the start of the treatment as well as after 48 h and 96 h.

Changes to the manuscript:

We have now updated the discussion (329-342) and methods (lines 371-372) section to include these details.

Reviewer comment 5:

5. Others have reported MHC upregulation at the transcript and protein level by DNMTi treatment, which would improve antigen presentation to T cells. Do the authors observe this in their treated cell lines?

Authors' response: We performed a differential expression analysis for all HLA transcripts in the cell line panel and did not observe a systematic induction of HLA transcripts upon Decitabine or SB939 treatment. Since we did not perform a FACs analysis, we cannot rule out an increased surface expression of HLA upon these treatments.

Changes to the manuscript:

The data is now included in the manuscript as Supplementary Figure 8g and the discussion section has been updated accordingly (lines 291-292).

Reviewer comment 6:

6. Figure 6 shows nicely that some of the LTR-derived peptides are presented on MHC of AML patients; this seems to occur both with and without DAC. Do these presented peptides activate T cells from the AML patients? This would prove an immune response in vivo.

Authors' response: Previously, Saini et al (Nat Communications, 2020) demonstrated that HERV-specific T cells are present in AML patients irrespective of DNMTi treatment. In line with these observations, we also detect novel ORF-derived peptides in pre- as well as post-DAC treatment in AML patients. However, the number of novel ORF-derived peptides was largely increased in samples after DAC treatment.

Immunogenicity testing of the TINPAT-derived peptides identified in the treated AML patients was not possible directly in the respective patients as patient material was very scarce and there was no possibility for further sample collection. However, we agree with the reviewer that immunogenicity testing of these antigens would provide a significant upgrade of our findings. Therefore, we selected nine HLA class II-restricted t-neopeptides identified in the immunopeptidomics analysis of the AML patients and tested these in Interferon- γ (IFN- γ) Enzyme-Linked Immunospot (ELISpot) assays using PBMCs of eight AML patients that had undergone DNMTi treatment to test for pre-existing peptide-specific CD4⁺ memory T cells. We could show t-neopeptide-specific CD4⁺ memory T cells in one out of the eight DNMTi-treated AML patients.

In addition, we analyzed three HLA class I t-neopeptides identified in the two AML patients for their ability to induce antigen-specific T cells using artificial antigen-presenting cell (aAPC)-based priming of CD8⁺ T cells of healthy volunteers (HV). We could show the induction of antigen-specific T cells against the A*03:01 ligand RTDSSLLEK.

Changes to the manuscript:

We now included new data showing the immunogenicity of t-neopeptides in the revised manuscript Figure 6f and Supplementary Figure 7a (lines 252-259).

We thank the reviewer for their valuable suggestions and hope that they are content with the newly added experiments, data, and information included in the revised manuscript and find the manuscript now acceptable for publication in Nature Communications.

Reviewers' Comments:

Reviewer #3:

Remarks to the Author:

The authors have fully responded to my questions with detailed answers and new information and I have no further questions. This is a comprehensive and clear study that significantly advances the field.

Reviewer #4:

Remarks to the Author:

The additional analyses added new and important findings to this manuscript. Although I think a more critical discussion of the large proportion of non-validated t-neopeptides after comparison with synthetic counterparts would have been necessary.

Overall, the authors have addressed the concerns sufficiently.

Reviewer #5:

Remarks to the Author:

My annotated reply to the authors is attached.

Reviewer #2, expertise in ERV-derived transcripts and bioinformatics

(Remarks to the Author):

Noteworthy results

The authors expand on their previous publication on treatment-induced novel-polyadenylated transcripts (Brock et al. Nature Genetics). Here, they use in silico predictions, immunopeptidomics, and AML patient-material after decitabine treatment, to characterise the potential of (mainly) LTR12-derived non-reference transcripts for immunogenic HLA-I epitopes. Importantly, the authors provide strong evidence for at least 15 HLA-I neo-epitopes upon DAC/SB393 treatment for cells with AML-origins.

Overall comment on publication

I expect the study will be of immediate interest to research into the consequences of decitabine / 5-azacitidine administration in AML, MDS and CML and physicians that consider combination and sequential treatment regimens with immunotherapy agents. It is also of interest to the growing field of long terminal repeats as alternative promoters.

The manuscript is well written and presented. It is warranted to address a limited number of concerns in the primary data analysis to support the conclusions, as listed below

Authors' response: We thank the reviewer for the positive feedback on our study. Please find below a detailed point-to-point response addressing the raised questions and concerns.

Reviewer's Concerns

Below a number of concerns on phrases and presentation of the data which should be addressed before publication.

Reviewer comment 1a:

In the context of immunotherapy, we need to ask of neo- or tumour-antigens:

- are they individualised or shared between patients
- is immunological tolerance and expression in normal tissue expected or not
- are they expressed and presented by a sufficiently large amount of the tumour cells (or by tumour stem cells, proliferating cells, etc.)

This is in particular relevant for neo-antigens discovered via peptidomics that are (partially) derived from genomics repeats

To this end, I would expect the authors to address the following:

(1a) peptide homology of t-neoepitopes

- Homology with other hypothetical peptides: a summary of a blast search (or similar) against a transcriptome reference (+ all TINATs) or the human genome; highlighting the potential of other loci to generate the neo-epitope in question (+/- 1 amino acid difference).

Context: The authors partially address this concern in Figure 3 and state that 25 out of 38 t-neoepitopes are derived from a single transcript and outside of the LTR12 sequences. However, the epitope in Figure 6f arises from the LTR12 sequence, and their example in Figure 3b shows an epitope mapped to a presumed unique locus extremely close to the annotated LTR12 element. Since the boundaries of retroelements in mammalian genomes are (i) not 100% precisely annotated and (ii) recombination events leading to stand alone LTR elements often involve incomplete matching of the repeat sequences and recombination with other repeat sequences, it seems very likely to me that some of these t-neoepitopes still derive from sequences with close homology across multiple loci in the human genome; which is an important piece of information in this study.

Authors' response: We thank the reviewer for their valid criticism. To address this issue, we took the 38 unique peptides (found in at least 2 of the 3 biological replicates) and investigated their occurrence in Uniprot as well as the predicted DAC+SB939 induced ORFs (from Figure 2a), allowing an increasing number of mismatches (0, 1, or 2). At 0 mismatches allowed, 25/38 (65%) t-neopeptides were uniquely mapped. After allowing for 1 mismatch, 23/38 (60%) of t-neopeptide still remained uniquely mapped. 16/38 (42%) t-neopeptides were uniquely mapped even after allowing for 2 mismatches indicating that these peptides are sufficiently divergent from known or TINPAT-derived proteins.

Changes to the manuscript:

This data is now included in the manuscript as Supplementary Figure 4c,d and Supplementary data 8. The manuscript text has been updated accordingly (lines 189-194)

While this criticism was addressed, I would not interpret this as adequately dealing with it. The critique is that "some" of the epitopes derive from sequences across the genome. As I understand it under the most stringent cutoff 35% are homologous to multiple genome regions and after 2 mismatches in a transcript of some length there are a majority that are multimapped. Is 2 mismatches a lot? What regions are they being mapped to? Are they known ERV homologs? For instance in Figure 6 the ORF is fairly long, the 2 mismatch cutoff seems fairly arbitrary, what fraction of the ORF length is it?

Reviewer comment 1b:

(1b) Homology between individuals: a summary of the polymorphism in the sequence of the genomic loci generating the amino acids of the t-neoepitopes shown in Table1 in the 1000 genome projects (or similar resource)

Authors' response: We thank the reviewer for this suggestion and have now performed an analysis of polymorphism at DNA as well as peptide level for the 15 t-neopeptides mentioned in Table 1. Overall, we found very low polymorphism at the loci encoding these peptides.

Changes to the manuscript:

The results are summarized in Supplementary data 9. 12/15 (80%) t-neopeptides had less than 0.001 % polymorphism. The relevant text in results (lines 202-205) and methods (lines 515-520) sections have been updated accordingly.

Are the other 20% highly polymorphic?

Reviewer comment 2

(2) Rephrase abstract

The abstract is attempting to draw attention by using big numbers a bit too freely.

-> e.g. 3,023 ERV-derived, treatment-induced novel polyadenylated transcripts (TINPATs), encoding for 61,426 open reading frames

The respective figure 2 also shows ~72,000 ORFs from known and annotated transcripts. In principle, the authors would be able to derive a false discovery rate for the 72,000 known ORFs to be truly translated, by asking how many have accumulated evidence for being translated across the scientific knowledgebase using a source of their choice (proteomics, ribosome sequencing, etc). I would pose that less than 10% of the 72,000 ORFs of known transcripts will have evidence of producing peptides. The likelihood of chimeric/non-chimeric transcripts without purifying selection pressure will be lower than that. I would find it more informative if the authors lead in the abstract with the number of spectra-validated peptides rather than the hypothetical number of ORFs.

In a similar direction, the final statement in the abstract aims to high:

Our findings highlight a novel mechanism of ERV-derived neoantigens in epigenetic and immune therapies

ERV-derived neoantigens have been hypothesized and demonstrated for decades, and regarding the potential for DNMT inhibitors to induce them, the author's own prior publication in Brock et al. states in the abstract:

The resulting transcripts frequently splice into protein-coding exons and encode truncated or chimeric ORFs translated into products with predicted abnormal or immunogenic functions.

Authors' response and changes to the manuscript: We have now modified the abstract to reflect the suggested changes by the reviewer (lines 33-45).

Thank you for addressing this concern.

Reviewer comment 3:

(3) missing item in discussion

The identification of >15 HLA-I tumour antigens is a major advance in the field.

Authors' response and changes to the manuscript: We agree with the reviewer on this point and have thus highlighted this in the discussion section of the revised manuscript (lines 295-299).

Thank you for addressing this concern.

Reviewer comment 4:

As part of the discussion, the authors lay out the idea of using such knowledge and the presence/diversity/abundance of t-neo-epitopes for engineering clinical response. They need to weigh the available evidence in favour but also the available evidence contradicting this idea. I found at least one study that did not find association with response for ERV expression levels after 5-azacytidine in MDS patients: Kazachenka Genome Medicine volume 11, Article number: 86 (2019) [link]. I suspect there are more of them across tumour indications.

Authors' response: We thank the reviewer for making this point. Ohtani et al. (Cancer research, 2020) demonstrated a strong association between LTR12C expression and patient response upon Azacytidine treatment. However, Kazachenka et al. (Genome Medicine, 2019) could not find such association in a larger cohort of patients treated with Azacytidine. Therefore, whether DAC monotherapy can induce t-neopeptide driven immune response remains unclear and it is likely that patients receiving DAC treatment might benefit from combinatorial therapies targeting t-neopeptides.

Changes to the manuscript: We have adapted the discussion section accordingly in the revised manuscript (lines 337-342).

Thank you for addressing this concern.

Reviewer comment 5:

(4) Expand and correct Figure 1/ Supp Figure 2e.

For the comparison of TINPAT expression upon treatment versus normal tissue, the expression of the chimeric TINPATs should be compared against the major annotated isoform(s) of the gene the TINPAT splices into. If for ~3000 TINPATs, >2100 are not at all expressed in a particular tissue, the boxplot in Figure S1e will average closely to 0 (as shown). If the authors would plot 2000 tissue-specific genes in the same manner, we would falsely conclude that all those genes are lowly expressed in human. This is harder to control for in non-chimeric novel transcripts, hence I suggest to present the two classes separately

Authors' response:

We thank the reviewer for pointing out this flaw in the analysis. Besides TINPATs, now we also plot the TPMs for transcripts that were annotated to the same gene locus as TINPATs but were classified as known (labeled as known isotypes).

Changes to the manuscript:

Supplementary Figure 2e has been updated accordingly.

Thank you for addressing this concern.

Reviewer comment 6:

Miscellaneous comments

sentence on line137: non-chimeric TINPATs were predicted to have low protein-coding potential ... could be explained by the smaller average length of non-chimeric TINPATs

The authors continue on line 151ff showing all possible ORFs from chimeric and non-chimeric TINPATs and Figure 2 a/b/c illustrate nicely that non-chimeric TINPATs indeed have comparably lower protein-

coding potential (smaller ORFs and less of them). The sentence at line137 is superfluous and somewhat confusing.

Authors' response and changes to the manuscript: We have now removed the following line from the main text: "However, this could be explained by the smaller average length of non-chimeric TINPATs, which is considered in the protein-coding potential calculation."

Thank you for addressing this concern.

We thank the reviewer for their valuable suggestions and hope that they are content with the newly added experiments, data, and information included in the revised manuscript and find the manuscript now acceptable for publication in Nature Communications.

Reviewer #3:

The authors have fully responded to my questions with detailed answers and new information and I have no further questions. This is a comprehensive and clear study that significantly advances the field.

Authors' response: We thank the reviewer for their positive evaluation. Their constructive criticism helped us to improve our study and bring it to its current state.

Reviewer #4, with expertise in immunopeptidomics to replace Reviewer #1:

The additional analyses added new and important findings to this manuscript. Although I think a more critical discussion of the large proportion of non-validated t-neopeptides after comparison with synthetic counterparts would have been necessary.

Overall, the authors have addressed the concerns sufficiently.

Authors' response and changes to the manuscript:

We thank the reviewer for their positive evaluation. We have now added a statement pertaining to the non-validated t-neopeptides to the discussion section of the revised manuscript (lines 303-313).

Reviewer #5, with expertise in ERV-derived transcripts, bioinformatics to replace Reviewer #2:

(Remarks to the Author, highlighted in Blue):

Reviewer #2, expertise in ERV-derived transcripts and bioinformatics

Reviewer's Concerns

Below a number of concerns on phrases and presentation of the data which should be addressed before publication.

Reviewer comment 1a:

In the context of immunotherapy, we need to ask of neo- or tumour-antigens:

- are they individualised or shared between patients
- is immunological tolerance and expression in normal tissue expected or not

- are they expressed and presented by a sufficiently large amount of the tumour cells (or by tumour stem cells, proliferating cells, etc.)

This is in particular relevant for neo-antigens discovered via peptidomics that are (partially) derived from genomics repeats

To this end, I would expect the authors to address the following:

(1a) peptide homology of t-neoepitopes

- Homology with other hypothetical peptides: a summary of a blast search (or similar) against a transcriptome reference (+ all TINATs) or the human genome; highlighting the potential of other loci to generate the neo-epitope in question (+/- 1 amino acid difference).

Context: The authors partially address this concern in Figure 3 and state that 25 out of 38 t-neoepitopes are derived from a single transcript and outside of the LTR12 sequences. However, the epitope in Figure 6f arises from the LTR12 sequence, and their example in Figure 3b shows an epitope mapped to a presumed unique locus extremely close to the annotated LTR12 element. Since the boundaries of retroelements in mammalian genomes are (i) not 100% precisely annotated and (ii) recombination events leading to stand alone LTR elements often involve incomplete matching of the repeat sequences and recombination with other repeat sequences, it seems very likely to me that some of these t-neoepitopes still derive from sequences with close homology across multiple loci in the human genome; which is an important piece of information in this study.

Authors' response: We thank the reviewer for their valid criticism. To address this issue, we took the 38 unique peptides (found in at least 2 of the 3 biological replicates) and investigated their occurrence in Uniprot as well as the predicted DAC+SB939 induced ORFs (from Figure 2a), allowing an increasing number of mismatches (0, 1, or 2). At 0 mismatches allowed, 25/38 (65%) t-neopeptides were uniquely mapped. After allowing for 1 mismatch, 23/38 (60%) of t-neopeptide still remained uniquely mapped. 16/38 (42%) t-neopeptides were uniquely mapped even after allowing for 2 mismatches indicating that these peptides are sufficiently divergent from known or TINPAT-derived proteins.

Changes to the manuscript:

This data is now included in the manuscript as Supplementary Figure 4c,d and Supplementary data 8. The manuscript text has been updated accordingly (lines 189-194)

While this criticism was addressed, I would not interpret this as adequately dealing with it. The critique is that "some" of the epitopes derive from sequences across the genome. As I understand it under the most stringent cutoff 35% are homologous to multiple genome regions and after 2 mismatches in a transcript of some length there are a majority that are multimapped. Is 2 mismatches a lot? What regions are they being mapped to? Are they known ERV homologs? For instance in Figure 6 the ORF is fairly long, the 2 mismatch cutoff seems fairly arbitrary, what fraction of the ORF length is it?

Authors' response and changes to the manuscript:

We thank the reviewer for the questions. We now provide this analysis allowing up to 3 mismatches (**Supplementary Figure 4c, 4d, 4e**). Since the t-neopeptides are on average ~9 amino acids long, 3 amino acids represent ~33% of the t-neopeptide. To demonstrate that this is a big divergence, we performed a similar analysis on a set of 29 SARS CoV-2-derived HLA Class I T cell epitopes (Table 2, Nelde et al, Nature Immunology 2020). These epitopes are of viral origin and therefore are not expected to be identified in the human ORFeome. We blasted the sequences of these 29 epitopes against UniProt or DAC+SB939-induced ORFs allowing for 0, 1, 2, or 3 mismatches. At 0 or 1 mismatches allowed, all of the 29 peptides had no matches to either UniProt or DAC+SB939 induced ORFs. However, after allowing for 2 mismatches, 8/29 (27%) of the peptides could be mapped to either UniProt or DAC+SB939-induced ORFs. After allowing for 3 mismatches, 26/29 (90%) of the peptides could be mapped to either UniProt or DAC+SB939-induced ORFs (**Supplementary Figure 4f, 4g, 4h**). These results demonstrate that allowing more than 2 mismatches in an epitope blast is impractical, as any sequence will eventually be mapped.

Regarding the regions where the multi-mapping peptides are mapped to, we now provide additional columns in Supplementary Data 8. The table now contains the transcript of origin of the epitope and ORF and the repeat annotation in case the corresponding transcription start site of the mapped transcript lies within a repeat. Expectedly, the majority of the multi mapping t-neopeptides are derived from LTR12-derived transcripts (**Supplementary Figure 4i, 4j**).

The relevant text has also been updated in the revised manuscript (lines 198-210).

Regarding the ORF in Figure 6, the ORF length is 181 amino acids and the t-neopeptides are 10 amino acids long, making the t-neopeptide 5.5% of the ORF.

Reviewer comment 1b:

(1b) Homology between individuals: a summary of the polymorphism in the sequence of the genomic loci generating the amino acids of the t-neopeptides shown in Table1 in the 1000 genome projects (or similar resource)

Authors' response: We thank the reviewer for this suggestion and have now performed an analysis of polymorphism at DNA as well as peptide level for the 15 t-neopeptides mentioned in Table 1. Overall, we found very low polymorphism at the loci encoding these peptides.

Changes to the manuscript:

The results are summarized in Supplementary data 9. 12/15 (80%) t-neopeptides had less than 0.001 % polymorphism. The relevant text in results (lines 202-205) and methods (lines 515-520) sections have been updated accordingly.

Are the other 20% highly polymorphic?

Authors' response and changes to the manuscript:

All of the t-neopeptides had less than 0.08 % polymorphism, which is quite low. The relevant text has been updated in the revised manuscript (line 213).

Reviewers' Comments:

Reviewer #5:

Remarks to the Author:

There seems to be some confusion as to the nature of the questions for which the authors are responding. The original reviewer asked very reasonable questions for which the authors did not directly reply. The reason for those questions is likely that we have now good evidence many ERVs are expressed in normal tissue. The authors suggest designing T cell therapies against ERVs, a set of comments which should be walked back for publication since there may be mapping to other regions of the genome its reasonable to ask how often this is the case.

First, the statement that viruses do not contain similar sequences in the genome is demonstrably untrue – there is a good degree of mimicry of the genome by viruses. The case of SCV2 is not apples to apples, this is a distraction and I would not include it. There are several papers on the surprising degree of homology between the self and viral proteome. Second, we are now treating ERVs as a repetitive set of self-proteins and asking the question of tolerance, as the original reviewer raised. The concern I raised is that when the authors say 65% of peptides are uniquely mapped the converse is that 35% are not uniquely mapped. I consider 35% to be a high number to then later suggest T cell therapies, and similarly for one mismatch. Mapping to other ERV loci is a concern that should be clearly addressed as the chances that loci may be expressed in a different normal tissue is not low.

For the second comment I had asked “Are the other 20% highly polymorphic?” – so are they? It’s a straightforward question.

The nature of both concerns is that the threshold for such things should be high. I am not just concerned about homology to genes – I am concerned about homology to other ERVs loci.

Finally showing stimulation of a peptide in HLA matched healthy volunteers that is expressed in a cancer patient is not evidence of tumor killing as stated in the abstract. Any such statements should be taken out. If they wanted to show evidence these were real antigens why not do a recall assay in those same patients? Ultimately this is a proof of concept paper about expression and presentation of ERV antigens. Its an advance and should be published, but several statements are made that are not supported and should be walked back.

Reviewer #6:

Remarks to the Author:

With additional information about peptide homology, the authors have sufficiently responded to the questions from Reviewer #5.

Point-by point reply:

Reviewer #5:

First, the statement that viruses do not contain similar sequences in the genome is demonstrably untrue – there is a good degree of mimicry of the genome by viruses.

Authors' response:

Previously, in our rebuttal letter we made the following statement: "These epitopes are of viral origin and therefore are not expected to be identified in the human ORFeome." This was not a part of the manuscript and we agree with the reviewer's assessment.

Changes to the manuscript (marked in yellow):

None

The case of SCV2 is not apples to apples, this is a distraction and I would not include it. There are several papers on the surprising degree of homology between the self and viral proteome.

Authors' response:

Using the Sarscov2 analysis we wanted to demonstrate to the reviewer that allowing for 3 mismatches is impractical for such an analysis. However, we agree that this is probably a distraction from our main message.

Changes to the manuscript (marked in yellow):

We have now removed the concerned statement and the associated analysis from the manuscript (Lines#204-211, Supplementary Figure 4 has been updated accordingly).

Second, we are now treating ERVs as a repetitive set of self-proteins and asking the question of tolerance, as the original reviewer raised. The concern I raised is that when the authors say 65% of peptides are uniquely mapped the converse is that 35% are not uniquely mapped. I consider 35% to be a high number to then later suggest T cell therapies, and similarly for one mismatch. Mapping to other ERV loci is a concern that should be clearly addressed as the chances that loci may be expressed in a different normal tissue is not low.

Authors' response:

We thank the reviewer for turning our attention to these not uniquely mapped t-neopeptides and their possible expression in normal tissues. We agree that these concerns are of importance and we want to highlight that we were not claiming that all of the identified t-neopeptides that we report on could be

targeted in a clinical setting. We modified this matter in the discussion part of the manuscript, where we highlighted that future studies have to evaluate the frequency and tumor/patient specificity of t-neopeptides to evaluate that they are safe. We now improved the clarity of this statement in the revised manuscript.

Changes to the manuscript (marked in yellow):

We adopted the discussion of the revised manuscript to better address the safety concerns of the reviewer (Line 382). The section now reads: “Future studies have to evaluate the frequency, and tumor/patient specificity, and the safety of t-neopeptides under DAC treatment in large cohorts as well as the clinical relevance of T cells targeting these antigens in vivo.”

For the second comment I had asked “Are the other 20% highly polymorphic?” – so are they? It’s a straightforward question.

The nature of both concerns is that the threshold for such things should be high. I am not just concerned about homology to genes – I am concerned about homology to other ERVs loci.

Authors’ response:

We apologize for the ambiguity. We found that all the t-neopeptides had less than 0.08% polymorphism, which is actually quite low.

Changes to the manuscript (marked in yellow):

We have now revised the statement in the results section to reflect this (Line#214). The section now reads: “All of the t-neopeptides had a very low sequence polymorphism (Supplementary data 9).”

Finally showing stimulation of a peptide in HLA matched healthy volunteers that is expressed in a cancer patient is not evidence of tumor killing as stated in the abstract. Any such statements should be taken out. If they wanted to show evidence these were real antigens why not do a recall assay in those same patients? Ultimately this is a proof of concept paper about expression and presentation of ERV antigens. Its an advance and should be published, but several statements are made that are not supported and should be walked back.

Authors’ response:

We thank the reviewer for highlighting this important point. Immunogenicity testing of the TINPAT-derived peptides identified in the treated AML patients was unfortunately not possible directly in the respective patients as patient material was very scarce and there was no possibility for further sample collection. Therefore we continued these important validation experiments with PBMCs of eight AML patients that

had undergone DNMTi treatment to test for pre-existing memory T cell responses. However, we agree with the reviewer that we can not make a statement about cancer cell killing and adopted this statement in the abstract of the revised manuscript.

Changes to the manuscript (marked in yellow):

We have now adopted the statement in the abstract of the revised manuscript (Line#44). The sentence reads: "We illustrate the potential of the identified t-neopeptides to elicit a T-cell response to effectively target cancer cells."

Reviewer #6: (Remarks to the author)

With additional information about peptide homology, the authors have sufficiently responded to the questions from Reviewer #5.

Authors' response:

We thank the reviewer very much for the review and the opinion that the revisions we made to the manuscript were satisfying.